# Smoothed Online Convex Optimization Based on Discounted-Normal-Predictor

**Lijun Zhang**[1,2], **Wei Jiang**[1], **Jinfeng Yi**[3], **Tianbao Yang**[4]

[1]National Key Laboratory for Novel Software Technology, Nanjing University, Nanjing, China
[2]Peng Cheng Laboratory, Shenzhen 518055, China
[3]JD AI Research, Beijing, China
[4]Department of Computer Science and Engineering, Texas A&M University, College Station, USA
{zhanglj, jiangw}@lamda.nju.edu.cn, yijinfeng@jd.com, tianbao-yang@tamu.edu

## Abstract

In this paper, we investigate an online prediction strategy named as Discounted-Normal-Predictor [Kapralov and Panigrahy, 2010] for smoothed online convex optimization (SOCO), in which the learner needs to minimize not only the hitting cost but also the switching cost. In the setting of learning with expert advice, Daniely and Mansour [2019] demonstrate that Discounted-Normal-Predictor can be utilized to yield nearly optimal regret bounds over any interval, even in the presence of switching costs. Inspired by their results, we develop a simple algorithm for SOCO: *Combining online gradient descent (OGD) with different step sizes sequentially by Discounted-Normal-Predictor*. Despite its simplicity, we prove that it is able to minimize the adaptive regret with switching cost, i.e., attaining nearly optimal regret with switching cost on every interval. By exploiting the theoretical guarantee of OGD for dynamic regret, we further show that the proposed algorithm can minimize the dynamic regret with switching cost in every interval.

## 1 Introduction

Recently, a variant of online convex optimization (OCO), named as smoothed OCO (SOCO) has received lots of attention in the machine learning community [Goel et al., 2019, Li and Li, 2020]. In each round $t$, the learner chooses an action $\mathbf{w}_t$ from a convex domain $\mathcal{W}$, and a convex loss function $f_t(\cdot) : \mathcal{W} \mapsto \mathbb{R}$ is selected by an adversary. While in the traditional OCO, the learner only suffers a *hitting* cost $f_t(\mathbf{w}_t)$, in SOCO, it further pays a *switching* cost $m(\mathbf{w}_t, \mathbf{w}_{t-1})$, which penalizes the learner for changing its actions between rounds. SOCO has found wide applications in real-world problems where the change of states brings additional costs [Lin et al., 2012, Kim and Giannakis, 2014, Goel et al., 2017], such as the wear-and-tear cost of switching servers [Lin et al., 2011].

For general convex functions, a natural choice of the switching cost is the distance between successive actions, i.e., $m(\mathbf{w}_t, \mathbf{w}_{t-1}) = \|\mathbf{w}_t - \mathbf{w}_{t-1}\|$. Then, the total loss in the $t$-th round becomes

$$f_t(\mathbf{w}_t) + \lambda G\|\mathbf{w}_t - \mathbf{w}_{t-1}\| \tag{1}$$

where $\lambda \geq 0$ is the trade-off parameter, and $G$ is the upper bound of the norm of gradients. Here, $G$ is introduced to ensure that the two terms weighted by $\lambda$ are on the same order. Following the convention of online learning [Cesa-Bianchi and Lugosi, 2006], we choose regret to measure the performance, and meanwhile take into account the switching cost. Let $T$ be the total number of iterations. The standard metric would be *regret with switching cost*:

$$\sum_{t=1}^{T} \left( f_t(\mathbf{w}_t) + \lambda G\|\mathbf{w}_t - \mathbf{w}_{t-1}\| \right) - \min_{\mathbf{w} \in \mathcal{W}} \sum_{t=1}^{T} f_t(\mathbf{w}) \tag{2}$$

36th Conference on Neural Information Processing Systems (NeurIPS 2022).

defined as the difference between the cumulative loss of the learner and that of the best fixed decision in hindsight. It is easy to verify that online gradient descent (OGD) [Zinkevich, 2003] attains an $O(\sqrt{(1+\lambda)T})$ bound for regret with switching cost (c.f. Theorem 6), which can be proved to be optimal [Gradu et al., 2020, Theorem 4].

However, regret is not suitable for changing environments in which the best decision may vary over time. To address this limitation, new performance measures, including adaptive regret and dynamic regret have been proposed [Zhang, 2020, Cesa-Bianchi and Orabona, 2021]. Adaptive regret enforces the algorithm to have a small regret over every interval [Hazan and Seshadhri, 2007, Daniely et al., 2015], which essentially measures the performance w.r.t. a changing comparator. By extending adaptive regret to SOCO, we obtain *adaptive regret with switching cost*:

$$\text{A-R-S}(T,\tau) = \max_{r \le T+1-\tau} \text{R-S}(r, r+\tau-1) \tag{3}$$

where $\tau$ is the length of the interval, and

$$\text{R-S}(r,s) = \sum_{t=r}^{s} \left( f_t(\mathbf{w}_t) + \lambda G \|\mathbf{w}_t - \mathbf{w}_{t-1}\| \right) - \min_{\mathbf{w} \in \mathcal{W}} \sum_{t=r}^{s} f_t(\mathbf{w})$$

is the regret with switching cost over interval $[r,s]$. To cope with changing environments, dynamic regret directly compares the learner against a sequence of comparators $\mathbf{u}_1, \ldots, \mathbf{u}_T \in \mathcal{W}$ [Zinkevich, 2003]. Similarly, we can incorporate the switching cost into dynamic regret, and obtain *dynamic regret with switching cost*:

$$\sum_{t=1}^{T} \left( f_t(\mathbf{w}_t) + \lambda G \|\mathbf{w}_t - \mathbf{w}_{t-1}\| \right) - \sum_{t=1}^{T} f_t(\mathbf{u}_t). \tag{4}$$

In the literature, there are only two works that investigate adaptive regret with switching cost. However, one of them relies on strong convexity [Gradu et al., 2020], and the other one makes use of sophisticated techniques [Zhang et al., 2022]. There also exist preliminary studies on dynamic regret with switching cost, but are limited to the special case that $\lambda = 1/G$ [Zhang et al., 2021a] or follow different settings [Li et al., 2018, Goel et al., 2019]. Besides, all of previous works only target one performance measure. Notice that for OCO, we do have algorithms that are able to minimize adaptive regret and dynamic regret simultaneously [Zhang et al., 2020, Cutkosky, 2020].

In this paper, we develop a simple algorithm for SOCO based on Discounted-Normal-Predictor [Kapralov and Panigrahy, 2010, 2011], which is designed for online bit prediction and can be used to combine two experts. In a recent study, Daniely and Mansour [2019] reveal that Discounted-Normal-Predictor automatically controls the switching cost, and with the help of projection, can be extended to support adaptive regret. Motivated by their observations, we first analyze a variant of Discounted-Normal-Predictor with conservative updating (abbr. DNP-cu), and prove that it suffers a small loss on every interval, even in the presence of switching costs. To ensure adaptivity, we choose conservative updating instead of projection because the former one can be analyzed more easily. Then, we create multiple OGD with different step sizes, and combine them sequentially by DNP-cu. Theoretical analysis shows that the proposed method achieves an $O(\sqrt{(1+\lambda)\tau \log T})$ bound for adaptive regret with switching cost. Furthermore, based on the fact that OGD is also equipped with dynamic regret bounds, we establish nearly optimal guarantees for dynamic regret with switching cost in every interval. Specifically, for any interval $[r,s]$ with length $\tau$, our method attains an $O(\sqrt{(1+\lambda)\tau(1+P_{r,s})\log T})$ bound for dynamic regret with switching cost, where

$$P_{r,s} = \sum_{t=r}^{s} \|\mathbf{u}_t - \mathbf{u}_{t+1}\| \tag{5}$$

is the path-length of an arbitrary comparator sequence $\mathbf{u}_r, \ldots, \mathbf{u}_s \in \mathcal{W}$.

Compared with state-of-the-art results [Zhang et al., 2021a, 2022], this paper has the following advantages.

1. Our $O(\sqrt{(1+\lambda)\tau \log T})$ adaptive regret bound is on the same order as that of Zhang et al. [2022], but our method is more simple. In contrast, their algorithm pieces together several techniques, including online coin betting, geometric covering intervals, and the reduction from unconstrained online learning to constrained online learning.

2. Although Zhang et al. [2021a] establish an $O(\sqrt{T(1 + P_{1,T})})$ dynamic regret bound with switching cost, they only consider the case that $\lambda = 1/G$ and the whole interval $[1, T]$. By comparison, our dynamic regret bound holds for any $\lambda \geq 0$ and any interval.

3. To the best of our knowledge, this is the first effort to minimize both adaptive regret and dynamic regret, under the setting of SOCO.

Finally, we would like to emphasize the strength of Discounted-Normal-Predictor, which, in our opinion, was not getting enough attention. The results of Daniely and Mansour [2019] and this paper demonstrate that Discounted-Normal-Predictor provides an elegant way (and a very different methodology) to minimize both adaptive regret and dynamic regret, with or without switching cost, under the setting of learning with expert advice (LEA) or OCO.

## 2 Related Work

We briefly review the related work on OCO and SOCO, as well as Discounted-Normal-Predictor.

### 2.1 Online Convex Optimization (OCO)

During the past decades, OCO has been extensively studied, and various algorithms have been proposed to minimize the regret, i.e., (2) with $\lambda = 0$ [Shalev-Shwartz, 2011, Hazan, 2016]. It is well-known that OGD [Zinkevich, 2003] achieves an $O(\sqrt{T})$ bound which is minimax optimal [Abernethy et al., 2008]. We can obtain tighter regret bounds if the loss function satisfies specifical curvature properties, such as strong convexity [Shalev-Shwartz et al., 2007], exponential concavity [Hazan et al., 2007, Orabona et al., 2012], and smoothness [Srebro et al., 2010, Chiang et al., 2012].

Adaptive regret has been explored under the setting of LEA [Littlestone and Warmuth, 1994, Freund et al., 1997, Adamskiy et al., 2012, György et al., 2012, Luo and Schapire, 2015] and OCO [Hazan and Seshadhri, 2007, 2009, Jun et al., 2017b, Zhang et al., 2019, 2021b, Wan et al., 2022]. We follow the definition of Daniely et al. [2015], which is given by (3) with $\lambda = 0$. For general convex functions, Jun et al. [2017a] establish an $O(\sqrt{\tau \log T})$ bound, which holds for any interval length $\tau$.

Dynamic regret is proposed by Zinkevich [2003], and its definition can be found by setting $\lambda = 0$ in (4). Existing works have analyzed two types of dynamic regret: (i) the general version where the comparator sequence is arbitrary [Hall and Willett, 2013, Zhang et al., 2018b, Zhao et al., 2020a, Baby and Wang, 2021], and (ii) the worst-case version where $\mathbf{u}_t$ is chosen as the minimizer of $f_t(\cdot)$ [Jadbabaie et al., 2015, Besbes et al., 2015, Yang et al., 2016, Mokhtari et al., 2016, Zhang et al., 2017, Zhao and Zhang, 2021]. For general convex functions, Ader attains an optimal $O(\sqrt{T(1 + P_{1,T})})$ bound, where $P_{1,T}$ is the path-length of $\mathbf{u}_1, \ldots, \mathbf{u}_T$ [Zhang et al., 2018b].

Both adaptive regret and dynamic regret are designed for changing environments, but our understanding of their relationship is quite limited. Currently, we know that it is possible to bound the worst-case dynamic regret by adaptive regret [Zhang et al., 2018c], and minimize adaptive regret and the general dynamic regret simultaneously [Zhang et al., 2020, Cutkosky, 2020].

### 2.2 Smoothed Online Convex Optimization (SOCO)

The research of SOCO is motivated by real-world applications where the switching cost plays a crucial role [Lin et al., 2011, 2012]. Besides regret, competitive ratio is another popular metric for SOCO, and a large number of algorithms have been proposed to yield dimension-free competitive ratio. However, all of them are limited to the *lookahead* setting where the learner can observe the hitting cost $f_t(\cdot)$ before deciding its action $\mathbf{w}_t$, and rely on strong conditions [Bansal et al., 2015, Antoniadis and Schewior, 2018, Chen et al., 2018, Goel et al., 2019, Goel and Wierman, 2019, Argue et al., 2020a]. We note that SOCO is closely related to convex body chasing (CBC) [Friedman and Linial, 1993, Antoniadis et al., 2016, Bansal et al., 2018, Argue et al., 2019, Bubeck et al., 2019, 2020, Argue et al., 2020b, Sellke, 2020]. For more details, please refer to Lin et al. [2020].

In the study of online control, Gradu et al. [2020] and Zhang et al. [2022] have developed adaptive algorithms for OCO with memory, which can be applied to bounding adaptive regret with switching cost. However, Gradu et al. [2020] need to assume the hitting cost is strongly convex. Although Zhang et al. [2022] successfully deliver an $O(\sqrt{(1 + \lambda)\tau \log T})$ bound, their method is rather complex.

---

**Algorithm 1** Discounted-Normal-Predictor

---
**Require:** Two parameters: $n$ and $Z$
 1: Set $x_1 = 0$, and $\rho = 1 - 1/n$
 2: **for** $t = 1, \ldots, T$ **do**
 3:    Predict $g(x_t)$
 4:    Receive $b_t$
 5:    Set $x_{t+1} = \rho x_t + b_t$
 6: **end for**

---

Specifically, they make use of online coin betting [Orabona and Pál, 2016], geometric covering intervals [Daniely et al., 2015], and the reduction from unconstrained online linear optimization to constrained online linear optimization [Cutkosky and Orabona, 2018].

For dynamic regret with switching cost, Zhang et al. [2021a] extend Ader to support switching costs, and prove an optimal $O(\sqrt{T(1 + P_{1,T})})$ bound, but they only consider the case that $\lambda = 1/G$. Other works on dynamic regret with switching cost are incomparable to our paper, because they either rely on strong convexity [Li et al., 2018, Li and Li, 2020], set the switching cost as the squared distance [Goel et al., 2019], or assume an upper bound of the total switching cost is given in advance [Chen et al., 2018, Zhao et al., 2020b].

### 2.3 Discounted-Normal-Predictor

Following the terminology of Kapralov and Panigrahy [2010], we introduce Discounted-Normal-Predictor in the context of the bit prediction problem. Let $b_1, \ldots, b_T$ be an adversarial sequence of bits, where $b_t \in [-1, 1]$ can take real values. In each round $t$, the algorithm is required to output a confidence level $c_t \in [-1, 1]$, then observes the value of $b_t$, and finally gets a *payoff* $c_t b_t$. The goal is to maximize the cumulative payoff of the algorithm $\sum_{t=1}^{T} c_t b_t$.

Let $n > 0$ be a parameter for the interval length, and define the discount factor $\rho = 1 - 1/n$. As the name suggests, Discounted-Normal-Predictor maintains a discounted deviation $x_t = \sum_{j=1}^{t-1} \rho^{t-1-j} b_j$ at each round $t$, and the prediction is determined by $g(x_t)$ for a confidence function $g(\cdot)$ defined as

$$g(x) = \text{sign}(x) \cdot \min \left( Z \cdot \text{erf} \left( \frac{|x|}{4\sqrt{n}} \right) e^{\frac{x^2}{16n}}, 1 \right) \tag{6}$$

where $Z > 0$ is a parameter, and $\text{erf}(x) = \frac{2}{\sqrt{\pi}} \int_0^x e^{-s^2} ds$ is the error function. The complete procedure is summarized in Algorithm 1. For any $Z \leq 1/e$, Kapralov and Panigrahy [2010, Theorem 14] have proved that Discounted-Normal-Predictor satisfies

$$\sum_{t=1}^{T} g(x_t) b_t \geq \max \left( \left| \sum_{j=1}^{T} b_j \right| - O\big(\sqrt{T \log(1/Z)}\big), -O\big(Z\sqrt{T}\big) \right)$$

where we set $n = T$ in Algorithm 1. By choosing $Z = o(1/T)$, we observe that it has $O(\sqrt{T \log T})$ regret against the strategy that predicts the majority bit (whose payoff is $|\sum_j b_j|$), as well as a subconstant $o(1)$ loss. To address the problem of learning with two experts, we can define $b_t$ as the difference between the losses of experts and restrict $c_t \in [0, 1]$. It follows that the algorithm suffers $O(\sqrt{T \log T})$ and $o(1)$ regret w.r.t. the two experts, respectively [Kapralov and Panigrahy, 2010, Lemma 15]. Discounted-Normal-Predictor can be applied to the general setting of $N$ experts, by aggregating experts one by one. We note that the problem of trading off regret to the best expert for regret to the "special" expert stems from the study of Even-Dar et al. [2007], and is later investigated by Sani et al. [2014].

Kapralov and Panigrahy [2010, Theorem 5] also investigate a variant of the adaptive regret, which uses an infinite window with geometrically decreasing weighting. To this end, they propose a conservative updating rule to control the value of the deviation $x_t$. Let $U(n) = O(\sqrt{n \log(1/Z)})$ be a constant such that $|g(x)| = 1$ for $|x| \geq U(n)$. The current bit $b_t$ is utilized to update $x_t$ only when the confidence of the algorithm is low i.e., $|x_t| < U(n)$ or when the algorithm predicts incorrectly i.e., $g(x_t) b_t < 0$. Otherwise, they will ignore $b_t$ and set $x_{t+1} = \rho x_t$. However, it remains open

whether their conservative updating works with the standard adaptive regret [Daniely et al., 2015], which is answered affirmatively by our Theorem 1.

Daniely and Mansour [2019] extend Discounted-Normal-Predictor to support the switching cost and the adaptive regret. The confidence function is modified slightly as[1]

$$g(x) = \Pi_{[0,1]}\left[\tilde{g}(x)\right] \tag{7}$$

where

$$\tilde{g}(x) = \sqrt{\frac{n}{8}}Z \cdot \mathrm{erf}\left(\frac{x}{\sqrt{8n}}\right)e^{\frac{x^2}{16n}}, \tag{8}$$

and $\Pi_{[0,1]}[\cdot]$ denotes the projection operation onto the set $[0,1]$. First, by a more careful analysis, they demonstrate that Discounted-Normal-Predictor has similar regret bounds even in the presence of switching costs. Second, they introduce a projection operation to prevent $x_t$ from being too large, and then derive tight bounds for the standard adaptive regret. Specifically, the updating rule for $x_t$ becomes

$$x_{t+1} = \Pi_{[-2,U(n)+2]}\left[\rho x_t + b_t\right] \tag{9}$$

where

$$U(n) = \tilde{g}^{-1}(1) \le \sqrt{16n \log \frac{1}{Z}}. \tag{10}$$

Our work is inspired by Daniely and Mansour [2019], but with the following differences.

- While Daniely and Mansour [2019] consider the setting of LEA, we investigate OCO.
- To bound the adaptive regret, Daniely and Mansour [2019] introduce the projection operation, as shown in (9). In contrast, we make use of the conservative updating [Kapralov and Panigrahy, 2010], and provide a more simple analysis.
- We study not only the adaptive regret, but also the dynamic regret. Our algorithm is equipped with nearly optimal bounds for both metrics, in the presence of switching costs.

## 3 Main Results

We take Discounted-Normal-Predictor with conservative updating (DNP-cu) as our meta-algorithm, and present its performance by taking the switching cost into consideration. Then, we use DNP-cu to combine multiple OGD sequentially, and show its adaptive regret and dynamic regret with switching cost. Due to space limitations, all the proofs are deferred to Appendix A.

### 3.1 The Meta-algorithm

To consist with previous studies, we describe DNP-cu from the perspective of bit prediction, and require the prediction to lie in $[0,1]$ so that it can be used later as a meta-algorithm to combine experts. The detailed procedure is summarized in Algorithm 2. Compared with the original algorithm [Kapralov and Panigrahy, 2010], we make two modifications.

1. We choose the confidence function $g(\cdot)$ in (7), whose property has been revealed by Daniely and Mansour [2019, Lemmas 18 and 19] more formally.
2. We adapt the updating rule to the fact that $g(\cdot)$ belongs to $[0,1]$ instead of $[-1,1]$. Specifically, we perform the standard updating in Line 6, when the confidence is low i.e., $x_t \in [0, U(n)]$, or when the algorithm predicts incorrectly i.e.,

$$x_t < 0 \& b_t > 0 \text{ or } x_t > U(n) \& b_t < 0.$$

   Otherwise, we follow Line 8, which only shrinks $\mathbf{x}_t$ and ignores $b_t$.

To analyze the performance of DNP-cu, we consider both the payoff and the switching cost, referred to as *reward* below. We have the following theorem which exploits the possibility that the magnitude of the bit sequence may be smaller than 1.

---

[1]Their definition of the error function drops the constant $\frac{2}{\sqrt{\pi}}$.

---
**Algorithm 2** Discounted-Normal-Predictor with conservative updating (DNP-cu)
---
**Require:** Two parameters: $n$ and $Z$
 1: Set $x_1 = 0$, $\rho = 1 - 1/n$, and $U(n) = \tilde{g}^{-1}(1)$
 2: **for** $t = 1, \dots, T$ **do**
 3:     Predict $g(x_t)$ where $g(\cdot)$ is defined in (7)
 4:     Receive $b_t$
 5:     **if** $x_t \in [0, U(n)]$ or $x_t < 0 \& b_t > 0$ or $x_t > U(n) \& b_t < 0$ **then**
 6:         Set $x_{t+1} = \rho x_t + b_t$
 7:     **else**
 8:         Set $x_{t+1} = \rho x_t$
 9:     **end if**
10: **end for**
---

**Theorem 1** *Suppose $Z \leq \frac{1}{e}$ and $n \geq \max\{8e, 16 \log \frac{1}{Z}\}$. For any bit sequence $b_1, \dots, b_T$ such that $|b_t| \leq \mu \leq 1$, the cumulative reward of Algorithm 2 over any interval $[r, s]$ with length $\tau$ satisfies*

$$
\begin{aligned}
\sum_{t=r}^{s} &\left( g(x_t)b_t - \frac{1}{\mu}|g(x_t) - g(x_{t+1})| \right) \\
&\geq \max\left( 0, \sum_{t=r}^{s} b_t - \frac{\tau}{n}(U(n) + 2\mu) - U(n) - \mu \right) - U(n) - \mu - Z\tau
\end{aligned}
\tag{11}
$$

*where $U(n)$ is defined in (10). Furthermore, for intervals starting from 1, we have*

$$
\sum_{t=1}^{s} \left( g(x_t)b_t - \frac{1}{\mu}|g(x_t) - g(x_{t+1})| \right) \geq \max\left( 0, \sum_{t=1}^{s} b_t - \frac{\tau(U(n) + 2\mu)}{n} - U(n) \right) - Z\tau. \tag{12}
$$

*And the change of successive predictions satisfies*

$$
|g(x_t) - g(x_{t+1})| \leq \mu\sqrt{\frac{1}{n}\log\frac{1}{Z}} + \frac{Z\mu}{4}. \tag{13}
$$

**Remark:** First, the above theorem reveals that DNP-cu controls the switching cost automatically, and the trade-off between the payoff and the switching cost is determined by the magnitude of the bit sequence. Second, thanks to the conservative updating, we are able to bound the cumulative reward over any interval, which can be exploited to support adaptive regret.

Recall that in the total loss (1), the trade-off parameter $\lambda$ could be arbitrary. As a result, to utilize DNP-cu for SOCO, we also need a flexible way to balance the payoff and the switching cost. To this end, we introduce $\lambda$ and define the reward in the $t$-round as $g(x_t)b_t - \lambda|g(x_t) - g(x_{t-1})|$. Then, our goal is to maximize

$$
\sum_{t=r}^{s} \left( g(x_t)b_t - \lambda|g(x_t) - g(x_{t-1})| \right) \tag{14}
$$

for every interval $[r, s]$.[2] Based on Theorem 1, a straightforward way is to multiply the bit sequence by $1/\lambda$, and then pass it to Algorithm 2. But in this way, the lower bound will scale linearly with $\lambda$. To improve the dependence on $\lambda$, we will multiply the bit sequence by $1/\sqrt{\lambda}$ [Daniely and Mansour, 2019], and have the following corollary based on (11) and (13) of Theorem 1.[3]

**Corollary 2** *Suppose $Z \leq \frac{1}{e}$, $n \geq \max\{8e, 16 \log \frac{1}{Z}\}$, and $b_1, \dots, b_T$ is a bit sequence such that $|b_t| \leq 1$. Running Algorithm 2 over the* scaled *bit sequence*

$$
\frac{b_1}{\max(\sqrt{\lambda}, 1)}, \dots, \frac{b_T}{\max(\sqrt{\lambda}, 1)},
$$

---

[2]We actually focus on $\sum_{t=r}^{s} \left( g(x_t)b_t - \lambda|g(x_t) - g(x_{t+1})| \right)$, which differs from (14) by a constant factor.
[3]We omit (12), because it is not used in the subsequent analysis.

*for any interval $[r, s]$ with length $\tau$, we have*

$$\sum_{t=r}^{s} \big(g(x_t)b_t - \lambda|g(x_t) - g(x_{t+1})|\big)$$

$$\geq \max\left(0, \sum_{t=r}^{s} b_t - \frac{\max(\sqrt{\lambda}, 1)U(n)(\tau + n)}{n} - \frac{2\tau + n}{n}\right) \tag{15}$$

$$- \max(\sqrt{\lambda}, 1)U(n) - 1 - \max(\sqrt{\lambda}, 1)Z\tau.$$

*And the change of successive predictions satisfies*

$$|g(x_t) - g(x_{t+1})| \leq \frac{1}{\max(\sqrt{\lambda}, 1)}\left(\sqrt{\frac{1}{n}\log\frac{1}{Z}} + \frac{Z}{4}\right). \tag{16}$$

**Remark:** From the definition of $U(n)$ in (10), we have

$$\sum_{t=r}^{s} \big(g(x_t)b_t - \lambda|g(x_t) - g(x_{t+1})|\big) \quad = \quad - O\left(\sqrt{(1+\lambda)n\log\frac{1}{Z}} + \sqrt{(1+\lambda)}Z\tau\right) \tag{17}$$

$$\overset{Z=O(1/T)}{=} - O\left(\sqrt{(1+\lambda)n\log T}\right).$$

for any interval $[r, s]$. Notice that the bound in (17) is independent of the interval length $\tau$, so it holds even when $\tau$ is *larger* than $n$. On the other hand, for any interval $[r, s]$ whose length is no larger than $n$, i.e., $\tau/n \leq 1$, we also have the following regret bound of Algorithm 2 w.r.t. the baseline strategy which always outputs 1:

$$\sum_{t=r}^{s} \big(g(x_t)b_t - \lambda|g(x_t) - g(x_{t+1})|\big) \overset{Z=O(1/T)}{=} \sum_{t=r}^{s} b_t - O\left(\sqrt{(1+\lambda)n\log T}\right). \tag{18}$$

As elaborated in the analysis (c.f. Lemma 3), when using Algorithm 2 to combine multiple algorithms, there are two issues that need to be addressed.

1. We do not destroy the theoretical guarantee of early algorithms, which is ensured by (17).
2. We can inherit the theoretical guarantee of the current algorithm, which is archived by (18).

### 3.2 Smoothed OGD

We introduce the following common assumptions for OCO [Shalev-Shwartz, 2011].

**Assumption 1** *All the functions $f_t$'s are convex over the domain $\mathcal{W}$.*

**Assumption 2** *The gradients of all functions are bounded by $G$, i.e.,*

$$\max_{\mathbf{w}\in\mathcal{W}} \|\nabla f_t(\mathbf{w})\| \leq G, \ \forall t \in [T]. \tag{19}$$

**Assumption 3** *The diameter of the domain $\mathcal{W}$ is bounded by $D$, i.e.,*

$$\max_{\mathbf{w},\mathbf{w}'\in\mathcal{W}} \|\mathbf{w} - \mathbf{w}'\| \leq D. \tag{20}$$

Without loss of generality, we assume

$$f_t(\mathbf{w}) \in [0, GD], \ \forall \mathbf{w} \in \mathcal{W}, \ t \in [T], \tag{21}$$

since we can always redefine $f_t(\mathbf{w})$ as $f_t(\mathbf{w}) - \min_{\mathbf{w}\in\mathcal{W}} f_t(\mathbf{w})$ which belongs to $[0, GD]$ according to (19) and (20).

First, we explain how to use DNP-cu to combine predictions of two algorithms designed for OCO. Let $\mathcal{A}^1$ and $\mathcal{A}^2$ be two online learners, and denote their predictions in the $t$-th round by $\mathbf{w}_t^1$ and $\mathbf{w}_t^2$, respectively. Let $\mathcal{A}$ be a meta-algorithm which outputs a convex combination of $\mathbf{w}_t^1$ and $\mathbf{w}_t^2$, i.e.,

$$\mathbf{w}_t = (1 - w_t)\mathbf{w}_t^1 + w_t\mathbf{w}_t^2 \tag{22}$$

where the weight $w_t \in [0, 1]$. We have the following lemma regarding the meta-regret of $\mathcal{A}$.

---
**Algorithm 3** Combiner
---
**Require:** Three parameters: $M$, $n$ and $Z$
**Require:** Two algorithms: $\mathcal{A}^1$ and $\mathcal{A}^2$
 1: Let $\mathcal{D}$ be an instance of DNP-cu, i.e., Algorithm 2, with parameter $n$ and $Z$
 2: Receive $\mathbf{w}_1^1$ and $\mathbf{w}_1^2$ from $\mathcal{A}^1$ and $\mathcal{A}^2$ respectively
 3: Receive the prediction $w_1$ from $\mathcal{D}$
 4: **for** $t = 1, \ldots, T$ **do**
 5:     Predict $\mathbf{w}_t$ according to (22)
 6:     Send the loss function $f_t(\cdot)$ to $\mathcal{A}^1$ and $\mathcal{A}^2$
 7:     Receive $\mathbf{w}_{t+1}^1$ and $\mathbf{w}_{t+1}^2$ from $\mathcal{A}^1$ and $\mathcal{A}^2$ respectively
 8:     Send the bit $\frac{\ell_t}{\max(\sqrt{\lambda},1)}$ to $\mathcal{D}$, where $\ell_t$ is defined in (28)
 9:     Receive the prediction $w_{t+1}$ from $\mathcal{D}$
10: **end for**
---

**Lemma 1** *Assume the outputs of $\mathcal{A}^1$ and $\mathcal{A}^2$ move slowly such that*

$$\|\mathbf{w}_t^1 - \mathbf{w}_{t+1}^1\| \le \frac{MD}{\lambda}, \text{ and } \|\mathbf{w}_t^2 - \mathbf{w}_{t+1}^2\| \le \frac{MD}{\lambda}, \ \forall t \in [T] \tag{23}$$

*where $M \ge 0$ is some constant. Under Assumptions 1, 2 and 3, the meta-regret of $\mathcal{A}$ w.r.t. $\mathcal{A}^1$ over any interval $[r, s]$ satisfies*

$$\sum_{t=r}^s \left(f_t(\mathbf{w}_t) + \lambda G\|\mathbf{w}_t - \mathbf{w}_{t+1}\|\right) - \sum_{t=r}^s \left(f_t(\mathbf{w}_t^1) + \lambda G\|\mathbf{w}_t^1 - \mathbf{w}_{t+1}^1\|\right)$$
$$\le -(1+M)GD \sum_{t=r}^s \left(w_t \ell_t - \lambda |w_t - w_{t+1}|\right) \tag{24}$$

*and the meta-regret of $\mathcal{A}$ w.r.t. $\mathcal{A}^2$ satisfies*

$$\sum_{t=r}^s \left(f_t(\mathbf{w}_t) + \lambda G\|\mathbf{w}_t - \mathbf{w}_{t+1}\|\right) - \sum_{t=r}^s \left(f_t(\mathbf{w}_t^2) + \lambda G\|\mathbf{w}_t^2 - \mathbf{w}_{t+1}^2\|\right)$$
$$\le -(1+M)GD \sum_{t=r}^s \left(w_t \ell_t - \lambda |w_t - w_{t+1}| - \ell_t\right) \tag{25}$$

*where*

$$\ell_t^1 = f_t(\mathbf{w}_t^1) + \lambda G\|\mathbf{w}_t^1 - \mathbf{w}_{t+1}^1\| \overset{(21),(23)}{\in} [0, (1+M)GD], \tag{26}$$

$$\ell_t^2 = f_t(\mathbf{w}_t^2) + \lambda G\|\mathbf{w}_t^2 - \mathbf{w}_{t+1}^2\| \overset{(21),(23)}{\in} [0, (1+M)GD], \tag{27}$$

$$\ell_t = \frac{\ell_t^1 - \ell_t^2}{(1+M)GD} \overset{(26),(27)}{\in} [-1, 1]. \tag{28}$$

**Remark:** By mapping $w_t$ and $\ell_t$ in Lemma 1 to $g(x_t)$ and $b_t$ in Corollary 2, we immediately see that the weight $w_t$ in (22) can be determined by invoking DNP-cu, i.e., Algorithm 2 to process the scaled bit sequence

$$\frac{\ell_1}{\max(\sqrt{\lambda}, 1)}, \ldots, \frac{\ell_T}{\max(\sqrt{\lambda}, 1)}.$$

Then, we can utilize lower bounds in Corollary 2 to establish upper bounds for the regret in Lemma 1. We name the strategy of aggregating two algorithms by DNP-cu as Combiner, and summarize its procedure in Algorithm 3.

In the following, we will use online gradient descent (OGD) with constant step size [Zinkevich, 2003] as our expert-algorithm. OGD performs gradient descent to update the current solution $\mathbf{w}_t$:

$$\mathbf{w}_{t+1} = \Pi_{\mathcal{W}} \left[\mathbf{w}_t - \eta \nabla f_t(\mathbf{w}_t)\right]$$

---

**Algorithm 4** Smoothed OGD

---

**Require:** Three parameters: $K$, $M$ and $Z$

1: **for** $i = 1, \ldots, K$ **do**
2:     Set $n^{(i)} = T2^{1-i}$
3:     Let $\mathcal{A}^i$ be an instance of OGD with step size $\eta^{(i)}$ defined in (29)
4:     **if** $i = 1$ **then**
5:         Set $\mathcal{B}^1 = \mathcal{A}^1$
6:     **else**
7:         Let $\mathcal{B}^i$ be an instance of Combiner, i.e., Algorithm 3 which combines $B^{i-1}$ and $\mathcal{A}^i$ with parameters $M$, $n^{(i)}$ and $Z$
8:     **end if**
9: **end for**
10: **for** $t = 1, \ldots, T$ **do**
11:     Run $\mathcal{B}^1, \ldots, \mathcal{B}^K$ sequentially for one step
12:     Predict the output of $\mathcal{B}^K$, denoted by $\mathbf{w}_t$
13: **end for**

---

where $\eta > 0$ is the step size, and $\Pi_{\mathcal{W}}[\cdot]$ denotes the projection onto $\mathcal{W}$. Notice that it is important to choose a *constant* step size, which makes it easy to analyze the regret, as well as the dynamic regret, over any interval $[r, s]$.

First, we create $K$ instances of OGD, denoted by $\mathcal{A}^1, \ldots, \mathcal{A}^K$, where the value of $K$ will be determined later. The step size of $\mathcal{A}^i$ is set to be

$$\eta^{(i)} = \frac{D}{G} \sqrt{\frac{1}{(1 + 2\lambda)n^{(i)}}} \tag{29}$$

where

$$n^{(i)} = T2^{1-i}. \tag{30}$$

Then, we use Combiner, i.e., Algorithm 3 to aggregate them sequentially. We will create a sequence of algorithms $\mathcal{B}^1, \ldots, \mathcal{B}^K$, where $\mathcal{B}^i$ is obtained by combining $\mathcal{B}^{i-1}$ with $\mathcal{A}^i$, and $\mathcal{B}^1 = \mathcal{A}^1$. The parameter $n$ in Algorithm 3 is set to be $n^{(i)}$ when forming $\mathcal{B}^i$. In each iteration $t$, we invoke $\mathcal{B}^1, \ldots, \mathcal{B}^K$ sequentially for one step, and return the output of $\mathcal{B}^K$ as the prediction $\mathbf{w}_t$. The completed procedure is named as smoothed OGD, and summarized in Algorithm 4.

**Remark:** Our method has a similar structure with those of Cutkosky [2020] and Zhang et al. [2022], in the sense that we all combine multiple experts sequentially. In contrast, other approaches for adaptive regret use a two-level framework, where a meta-algorithm aggregates multiple experts (which is allowed to sleep) simultaneously [Hazan and Seshadhri, 2007, Daniely et al., 2015, Jun et al., 2017a,b, Zhang et al., 2019, 2021b]. On the other hand, all the previous works, including Cutkosky [2020] and Zhang et al. [2022], need to construct a set of sub-intervals, and maintain a sub-routine for each one. In this way, they can use a small number of sub-intervals to cover any possible interval, and attain a small regret on that interval. By comparison, our method does not rely on any special construction of sub-intervals, making it more elegant.

### 3.3 Theoretical Guarantees

Next, we provide the theoretical guarantee of smoothed OGD. We first characterize its regret with switching cost over any interval.

**Theorem 3** *Assume*

$$T \geq \max(\sqrt{\lambda} \log_2 T, e) \tag{31}$$

*and set*

$$K = \left\lfloor \log_2 \frac{T}{32 \max(\lambda, 1) \log 1/Z} \right\rfloor + 1, \tag{32}$$

*M = 2 and Z = 1/T in Algorithm 4. Under Assumptions 1, 2 and 3, we have*

$$\sum_{t=r}^{s} \left( f_t(\mathbf{w}_t) + \lambda G \|\mathbf{w}_t - \mathbf{w}_{t+1}\| - f_t(\mathbf{w}) \right)$$

$$\leq 2GD\sqrt{(1+\lambda)\tau} + 113GD \max(\sqrt{\lambda}, 1)\sqrt{\tau \log T} = O\left( \sqrt{(1+\lambda)\tau \log T} \right)$$

*for any interval $[r, s]$ with length $\tau$, and any $\mathbf{w} \in \mathcal{W}$.*

**Remark:** Our $O(\sqrt{(1+\lambda)\tau \log T})$ bound is on the same order as that of Zhang et al. [2022]. When $\lambda = 0$, we get the $O(\sqrt{\tau \log T})$ adaptive regret for general convex functions [Jun et al., 2017a].

Our proposed method is also equipped with nearly optimal dynamic regret with switching cost over any interval, as stated below.

**Theorem 4** *Under the condition of Theorem 3, we have*

$$\sum_{t=r}^{s} \left( f_t(\mathbf{w}_t) + \lambda G \|\mathbf{w}_t - \mathbf{w}_{t+1}\| - f_t(\mathbf{u}_t) \right)$$

$$\leq 2GD\sqrt{(1+\lambda)\tau(1 + 2P_{r,s}/D)} + 120GD \max(\sqrt{\lambda}, 1)\sqrt{\tau(1 + 2P_{r,s}/D) \log T}$$

$$= O\left( \sqrt{(1+\lambda)\tau(1 + P_{r,s}) \log T} \right)$$

*where $\tau$ is the interval length, and $P_{r,s}$, defined in (5), is the path-length of an arbitrary comparator sequence $\mathbf{u}_r, \ldots, \mathbf{u}_s \in \mathcal{W}$.*

**Remark:** The above theorem shows that our method can minimize the dynamic regret with switching cost over any interval. According to the $\Omega(\sqrt{T(1 + P_{1,T})})$ lower bound of dynamic regret [Zhang et al., 2018b, Theorem 2], we know that our upper bound is optimal, up to a logarithmic factor. Theorem 4 is very general and can be simplified in different ways.

1. If we choose a fixed comparator such that $P_{r,s} = 0$, Theorem 4 reduces to Theorem 3 and matches that of Zhang et al. [2022].
2. If we ignore the switching cost and set $\lambda = 0$, we obtain an $O(\sqrt{\tau(1 + P_{r,s}) \log T})$ bound for dynamic regret over any interval, which recovers the results of Zhang et al. [2020, Theorem 4] and Cutkosky [2020, Theorem 7].
3. When both $P_{r,s} = 0$ and $\lambda = 0$, we obtain the $O(\sqrt{\tau \log T})$ adaptive regret of Jun et al. [2017a].

## 4 Conclusion and Future Work

Based on a variant of Discounted-Normal-Predictor (DNP-cu), we design a novel algorithm, named as smoothed OGD for SOCO. Our algorithm combines multiple instances of OGD sequentially by DNP-cu, and thus is very simple. Theoretical analysis shows that it attains nearly optimal bounds for both adaptive regret and dynamic regret over any interval, in the presence of switching costs. We also conduct preliminary experiments to verify our theories, and present the results in Appendix B.

For adaptive regret and dynamic regret, we can obtain tighter bounds when the hitting cost exhibits additional curvature properties such as exponential concavity [Hazan and Seshadhri, 2007, Baby and Wang, 2021] and smoothness [Zhang et al., 2019, Zhao et al., 2020a]. It remains unclear whether DNP-cu can exploit such information to further improve the performance. For SOCO, it is common to consider the lookahead setting, and the problem is still nontrivial due to the coupling created by the switching cost [Chen et al., 2018]. For dynamic regret with switching cost, Zhang et al. [2021a] have demonstrated that Assumption 2 is unnecessary in the lookahead setting. It would be interesting to develop a lookahead version of DNP-cu, and investigate whether we can drop Assumption 2 as well.

## Acknowledgments and Disclosure of Funding

This work was partially supported by NSFC (62122037, 61976112), and the Major Key Project of PCL (PCL2021A12).

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
