# A   Analysis

In this section, we present the proof of all theorems.

## A.1   Proof of Theorem 1

First, from the updating rule in Algorithm 2, we can prove that the derivation satisfies

$$-\mu \leq x_t \leq U(n) + \mu, \ \forall t \geq 1. \tag{33}$$

To see this, we first consider the upper bound in (33). Let $k$ be any iteration such that $x_k \leq U(n)$ and $x_{k+1} > U(n)$. Then, we must have $x_{k+1} = \rho x_k + b_k$, because otherwise $x_{k+1} = \rho x_k < U(n)$. As a result,

$$x_{k+1} = \rho x_k + b_k \leq U(n) + \mu.$$

Now, we consider the next derivation $x_{k+2}$. Because $x_{k+1} > U(n)$, according to the conservative updating rule, we have

$$x_{k+2} = \begin{cases} \rho x_{k+1} + b_{k+1}, & b_{k+1} < 0; \\ \rho x_{k+1}, & \text{otherwise.} \end{cases}$$

which is always smaller than $x_{k+1}$. Repeating the above argument, we conclude that the subsequent derivations $x_{k+2}, x_{k+3}, \ldots$ keep decreasing until they become no bigger than $U(n)$. As a result, it is impossible for $x_t$ to exceed $U(n) + \mu$.

The lower bound in (33) can be proved in a similar way. Let $k$ be any iteration such that $x_k \geq 0$ and $x_{k+1} < 0$. Then, we must have $x_{k+1} = \rho x_k + b_k$, because otherwise $x_{k+1} = \rho x_k \geq 0$. As a result,

$$x_{k+1} = \rho x_k + b_k \geq b_k \geq -\mu.$$

Now, we consider the next derivation $x_{k+2}$. Because $x_{k+1} < 0$, according to the conservative updating rule, we have

$$x_{k+2} = \begin{cases} \rho x_{k+1} + b_{k+1}, & b_{k+1} > 0; \\ \rho x_{k+1}, & \text{otherwise.} \end{cases}$$

which is always bigger than $x_{k+1}$. Repeating the above argument, we conclude that the subsequent derivations $x_{k+2}, x_{k+3}, \ldots$ keep increasing until they become nonnegative. As a result, it is impossible for $x_t$ to be smaller than $-\mu$.

Next, we make use of Algorithm 1 to analyze the reward of Algorithm 2. Following Kapralov and Panigrahy [2010], we construct the following bit sequence

$$\tilde{b}_t = \begin{cases} b_t, & \text{if Line 6 of Algorithm 2 is executed at round } t; \\ 0, & \text{otherwise.} \end{cases}$$

It is easy to verify that the prediction $g(x_t)$, as well as the derivation $x_t$, of Algorithm 2 over the bit sequence $b_1, \ldots, b_T$ is exact the same as that of Algorithm 1 over the new sequence $\tilde{b}_1, \ldots, \tilde{b}_T$. Since Algorithm 1 is more simple, we will first establish the theoretical guarantee of Algorithm 1 over the new sequence, and then convert it to the reward of Algorithm 2 over the original sequence. We have the following theorem for Algorithm 1 [Daniely and Mansour, 2019].

**Theorem 5** *Suppose $Z \leq \frac{1}{e}$ and $n \geq \max\{8e, 16 \log \frac{1}{Z}\}$. For any bit sequence $b_1, \ldots, b_T$ such that $|b_t| \leq \mu \leq 1$, the cumulative reward of Algorithm 1 over any interval $[r, s]$ with length $\tau$ satisfies*

$$\sum_{t=r}^{s} \left( g(x_t)b_t - \frac{1}{\mu}|g(x_t) - g(x_{t+1})| \right)$$

$$\geq \max\left( 0, \sum_{t=r}^{s} b_t + x_r - \frac{\tau}{n}\left( U(n) + 2\mu \right) - U(n) \right) - \max(x_r, 0) - Z\tau \tag{34}$$

*where $U(n)$ is defined in (10). And the change of successive predictions satisfies*

$$|g(x_t) - g(x_{t+1})| \leq \mu\sqrt{\frac{1}{n}\log\frac{1}{Z}} + \frac{Z\mu}{4}. \tag{35}$$

Theorem 5 can be extracted from the proofs of Lemmas 21 and 23 of Daniely and Mansour [2019]. For the sake of completeness, we provide its analysis in Appendix A.10. We can see that the lower bound in (34) depends on $x_r$, which explains the necessity of controlling its value.

Notice that $\mu$ is also the upper bound of the absolute value of the new sequence $\tilde{b}_1, \ldots, \tilde{b}_T$. According to Theorem 5, we directly obtain (13) from (35). From (34), we have

$$
\begin{aligned}
&\sum_{t=r}^{s} \left( g(x_t)\tilde{b}_t - \frac{1}{\mu}|g(x_t) - g(x_{t+1})| \right) \\
&\geq \max\left( 0, \sum_{t=r}^{s} \tilde{b}_t + x_r - \frac{\tau}{n}\left( U(n) + 2\mu \right) - U(n) \right) - \max(x_r, 0) - Z\tau.
\end{aligned}
\tag{36}
$$

On the other hand, the reward in terms of the original sequence is

$$
\begin{aligned}
&\sum_{t=r}^{s} \left( g(x_t)b_t - \frac{1}{\mu}|g(x_t) - g(x_{t+1})| \right) \\
&= \sum_{t=r}^{s} g(x_t)(b_t - \tilde{b}_t) + \sum_{t=r}^{s} \left( g(x_t)\tilde{b}_t - \frac{1}{\mu}|g(x_t) - g(x_{t+1})| \right).
\end{aligned}
\tag{37}
$$

So, we need to bound $\sum_{t=r}^{s} g(x_t)(b_t - \tilde{b}_t)$. Let $k$ be any iteration such that $b_k \neq \tilde{b}_k$, i.e., Line 8 of Algorithm 2 is executed at round $k$, which also implies $\tilde{b}_k = 0$. From the updating rule, we must have

$$
x_k < 0 \,\&\, b_k \leq 0 \text{ or } x_k > U(n) \,\&\, b_k \geq 0.
$$

If $x_k < 0 \,\&\, b_k \leq 0$, we have

$$
g(x_k)(b_k - \tilde{b}_k) = g(x_k)b_k = 0 \geq b_k = b_k - \tilde{b}_k
$$

since $g(x_k) = 0$ and $\tilde{b}_k = 0$. Otherwise if $x_k > U(n) \,\&\, b_k \geq 0$, we have

$$
g(x_k)(b_k - \tilde{b}_k) = b_k = b_k - \tilde{b}_k \geq 0
$$

since $g(x_k) = 1$ and $\tilde{b}_k = 0$. So, we always have

$$
g(x_k)(b_k - \tilde{b}_k) \geq \max\left( 0, b_k - \tilde{b}_k \right), \text{ if } b_k \neq \tilde{b}_k.
\tag{38}
$$

As a result,

$$
\begin{aligned}
\sum_{t=r}^{s} g(x_t)(b_t - \tilde{b}_t) &= \sum_{t\in[r,s]\,\&\,b_t \neq \tilde{b}_t} g(x_t)(b_t - \tilde{b}_t) \\
&\overset{(38)}{\geq} \max\left( 0, \sum_{t\in[r,s]\,\&\,b_t \neq \tilde{b}_t} \left( b_t - \tilde{b}_t \right) \right) = \max\left( 0, \sum_{t=r}^{s} \left( b_t - \tilde{b}_t \right) \right).
\end{aligned}
\tag{39}
$$

Combining (36), (37) and (39), we have

$$
\begin{aligned}
&\sum_{t=r}^{s} \left( g(x_t)b_t - \frac{1}{\mu}|g(x_t) - g(x_{t+1})| \right) \\
&\geq \max\left( 0, \sum_{t=r}^{s} \tilde{b}_t + x_r - \frac{\tau}{n}\left( U(n) + 2\mu \right) - U(n) \right) - \max(x_r, 0) - Z\tau \\
&\quad + \max\left( 0, \sum_{t=r}^{s} \left( b_t - \tilde{b}_t \right) \right) \\
&\geq \max\left( 0, \sum_{t=r}^{s} b_t + x_r - \frac{\tau}{n}\left( U(n) + 2\mu \right) - U(n) \right) - \max(x_r, 0) - Z\tau.
\end{aligned}
$$

Then, we obtain (11) by using (33) to bound $x_r$, and obtain (12) based on $x_1 = 0$.

## A.2 Proof of Corollary 2

Notice that the magnitude of the scaled bit sequence is upper bounded by $\mu = 1/\max(\sqrt{\lambda}, 1)$. From Theorem 1, we have

$$\sum_{t=r}^{s} \left( g(x_t) \frac{b_t}{\max(\sqrt{\lambda}, 1)} - \max(\sqrt{\lambda}, 1)|g(x_t) - g(x_{t+1})| \right)$$

$$\overset{(11)}{\geq} \max\left( 0, \sum_{t=r}^{s} \frac{b_t}{\max(\sqrt{\lambda}, 1)} - \frac{\tau}{n}\left( U(n) + \frac{2}{\max(\sqrt{\lambda}, 1)} \right) - U(n) - \frac{1}{\max(\sqrt{\lambda}, 1)} \right) \quad (40)$$

$$- U(n) - \frac{1}{\max(\sqrt{\lambda}, 1)} - Z\tau.$$

Then, we can lower bound the cumulative reward as follows

$$\sum_{t=r}^{s} \left( g(x_t)b_t - \lambda|g(x_t) - g(x_{t-1})| \right)$$

$$\geq \max(\sqrt{\lambda}, 1) \sum_{t=r}^{s} \left( g(x_t) \frac{b_t}{\max(\sqrt{\lambda}, 1)} - \max(\sqrt{\lambda}, 1)|g(x_t) - g(x_{t+1})| \right)$$

$$\overset{(40)}{\geq} \max\left( 0, \sum_{t=r}^{s} b_t - \max(\sqrt{\lambda}, 1)U(n)\left( \frac{\tau}{n} + 1 \right) - \frac{2\tau}{n} - 1 \right) - \max(\sqrt{\lambda}, 1)U(n)$$

$$- 1 - \max(\sqrt{\lambda}, 1)Z\tau$$

which proves (15). The upper bound in (16) is a direct consequence of (13).

## A.3 Proof of Theorem 3

First, we show that under our setting of parameters, all the preconditions in Corollary 2 and Lemma 1 are satisfied so that they can be exploited to analyze $\mathcal{B}^i$, which invokes Algorithm 2 to combine $\mathcal{B}^{i-1}$ and $\mathcal{A}^i$. From (31), we know that $Z = 1/T \leq 1/e$. From our definition of $K$, we have

$$n^{(i)} \geq T2^{1-K} \overset{(32)}{\geq} 32\max(\lambda, 1)\log \frac{1}{Z} \geq 32\log \frac{1}{Z} \geq 32 \geq 8e, \ \forall i \in [K]. \quad (41)$$

Thus, the conditions about $Z$ and $n$ in Corollary 2 are satisfied. Furthermore, our choice of $M$ ensures that (23) in Lemma 1 is true. To this end, we prove the following lemma.

**Lemma 2** *For all $\mathcal{A}^i$'s and $\mathcal{B}^i$'s created in Algorithm 4, their outputs satisfy the condition in (23) with $M = 2$.*

Based on above discussions, we conclude that Corollary 2 and Lemma 1 can be used in our analysis.

Next, we introduce the following theorem about the regret of OGD with switching cost over any interval $[r, s]$, which will be used to analyze the performance of $\mathcal{A}^i$'s.

**Theorem 6** *Let $\mathbf{w}_t$ be the outputs of OGD with step size $\eta$. Under Assumptions 1, 2 and 3, we have*

$$\sum_{t=r}^{s} \left( f_t(\mathbf{w}_t) + \lambda G\|\mathbf{w}_t - \mathbf{w}_{t+1}\| - f_t(\mathbf{w}) \right) \leq \frac{D^2}{2\eta} + \frac{(1+2\lambda)\eta(s-r+1)G^2}{2}$$

*for any $\mathbf{w} \in \mathcal{W}$.*

**Long Intervals** We proceed to analyze the performance of Algorithm 4 over an interval $[r, s]$, and start with the case that the interval length

$$\tau = s - r + 1 \geq 32\max(\lambda, 1)\log \frac{1}{Z}.$$

From our construction of $n^{(i)}$ in (30), there must exist a

$$k = \left\lfloor \log_2 \frac{T}{\tau} \right\rfloor + 1 \leq K \quad (42)$$

such that

$$\frac{n^{(k)}}{2} \leq \tau \leq n^{(k)}. \tag{43}$$

Then, we divide the proof into two steps:

(i) We show that the algorithm $\mathcal{A}^k$ attains an optimal regret with switching cost over the interval $[r, s]$;

(ii) We demonstrate that the regret of $\mathcal{B}^K$ w.r.t. $\mathcal{A}^k$ is under control.

Let $\mathbf{w}_t^k$ be the output of $\mathcal{A}^k$ in the $t$-th iteration. From Theorem 6, we have

$$\sum_{t=r}^{s} \left( f_t(\mathbf{w}_t^k) + \lambda G \|\mathbf{w}_t^k - \mathbf{w}_{t+1}^k\| - f_t(\mathbf{w}) \right)$$

$$\leq \frac{D^2}{2\eta^{(k)}} + \frac{(1+2\lambda)\eta^{(k)}\tau G^2}{2} \overset{(29)}{=} \frac{GD}{2}\sqrt{(1+2\lambda)n^{(k)}} + \frac{GD}{2}\tau\sqrt{\frac{1+2\lambda}{n^{(k)}}} \tag{44}$$

$$\overset{(43)}{\leq} \frac{(\sqrt{2}+1)GD}{2}\sqrt{(1+2\lambda)\tau} \leq 2GD\sqrt{(1+\lambda)\tau}.$$

Let $\mathbf{v}_t^i$ be the output of $\mathcal{B}^i$ in the $t$-th iteration. We establish the following lemma to bound the regret of $\mathcal{B}^K$ w.r.t. $\mathcal{A}^k$.

**Lemma 3** *For any interval $[r, s]$ with length $\tau \leq cn^{(k)}$, we have*

$$\sum_{t=r}^{s} \left( f_t(\mathbf{v}_t^K) + \lambda G \|\mathbf{v}_t^K - \mathbf{v}_{t+1}^K\| \right) - \sum_{t=r}^{s} \left( f_t(\mathbf{w}_t^k) + \lambda G \|\mathbf{w}_t^k - \mathbf{w}_{t+1}^k\| \right)$$

$$\leq GD \max(\sqrt{\lambda}, 1) \left( (12c + 53)\sqrt{n^{(k)} \log T} + 9 + 6c + 6(K - k) \right). \tag{45}$$

Based on Lemma 3, we have

$$\sum_{t=r}^{s} \left( f_t(\mathbf{v}_t^K) + \lambda G \|\mathbf{v}_t^K - \mathbf{v}_{t+1}^K\| \right) - \sum_{t=r}^{s} \left( f_t(\mathbf{w}_t^k) + \lambda G \|\mathbf{w}_t^k - \mathbf{w}_{t+1}^k\| \right)$$

$$\overset{(43),(45)}{\leq} GD \max(\sqrt{\lambda}, 1) \left( 65\sqrt{2\tau \log T} + 15 \right) + 6GD \max(\sqrt{\lambda}, 1)(K - k) \tag{46}$$

$$\leq 107 GD \max(\sqrt{\lambda}, 1)\sqrt{\tau \log T} + 6GD \max(\sqrt{\lambda}, 1)\log_2 \tau$$

$$\overset{\log_2 \tau \leq \sqrt{\tau \log \tau}}{\leq} 113 GD \max(\sqrt{\lambda}, 1)\sqrt{\tau \log T}$$

where in the penultimate step we make use of the following fact

$$K - k \overset{(32),(42)}{=} \left\lfloor \log_2 \frac{T}{32 \max(\lambda, 1) \log 1/Z} \right\rfloor - \left\lfloor \log_2 \frac{T}{\tau} \right\rfloor$$

$$\leq \log_2 \frac{\tau}{32 \max(\lambda, 1) \log 1/Z} + 1 \leq \log_2 \tau.$$

Combining (44) and (46), we have

$$\sum_{t=r}^{s} \left( f_t(\mathbf{w}_t^K) + \lambda G \|\mathbf{w}_t^K - \mathbf{w}_{t+1}^K\| - f_t(\mathbf{w}) \right)$$

$$\leq 2GD\sqrt{(1+\lambda)\tau} + 113 GD \max(\sqrt{\lambda}, 1)\sqrt{\tau \log T}. \tag{47}$$

**Short Intervals** We study short intervals $[r, s]$ such that

$$\tau = s - r + 1 \leq 32 \max(\lambda, 1) \log \frac{1}{Z}.$$

From Lemma 2, we know that the output of $\mathcal{B}^K$ moves slowly such that

$$\|\mathbf{w}_t^K - \mathbf{w}_{t+1}^K\| \leq \frac{2D}{\lambda}. \tag{48}$$

As a result, the regret of $B^K$ over $[r, s]$ can be bounded by

$$\sum_{t=r}^{s} \left( f_t(\mathbf{w}_t^K) + \lambda G \| \mathbf{w}_t^K - \mathbf{w}_{t+1}^K \| - f_t(\mathbf{w}) \right) \leq \sum_{t=r}^{s} \left( f_t(\mathbf{w}_t^K) + \lambda G \| \mathbf{w}_t^K - \mathbf{w}_{t+1}^K \| \right)$$

(49)

$$\overset{(21),(48)}{\leq} 3\tau GD \leq 3GD\sqrt{\tau \cdot 32 \max(\lambda, 1) \log T} \leq 17GD \max(\sqrt{\lambda}, 1)\sqrt{\tau \log T}.$$

We complete the proof by combing (47) and (49).

## A.4 Proof of Theorem 4

Since we focus on dynamic regret, so we need the following theorem regarding the dynamic regret of OGD with switching cost over any interval $[r, s]$.

**Theorem 7** *Under Assumptions 1, 2 and 3, we have*

$$\sum_{t=r}^{s} \left( f_t(\mathbf{w}_t) + \lambda G \| \mathbf{w}_t - \mathbf{w}_{t+1} \| - f_t(\mathbf{u}_t) \right) \leq \frac{D^2}{2\eta} + \frac{D}{\eta} \sum_{t=r}^{s} \| \mathbf{u}_t - \mathbf{u}_{t+1} \| + \frac{(1 + 2\lambda)\eta(s - r + 1)G^2}{2}$$

*for any comparator sequence* $\mathbf{u}_r, \ldots, \mathbf{u}_s \in \mathcal{W}$.

The proof is similar to that of Theorem 3, and we consider two scenarios: long intervals and short intervals. Here, we multiply the interval length $\tau$ by $1/(1 + 2P_{r,s}/D)$ to reflect the fact that the comparator is changing.

**Long Intervals** First, we study the case that

$$\frac{\tau}{1 + 2P_{r,s}/D} \geq 32 \max(\lambda, 1) \log \frac{1}{Z}.$$

From our construction of $n^{(i)}$ in (30), there must exist a

$$k = \left\lfloor \log_2 \frac{T(1 + 2P_{r,s}/D)}{\tau} \right\rfloor + 1 \leq K$$

(50)

such that

$$\frac{n^{(k)}}{2} \leq \frac{\tau}{1 + 2P_{r,s}/D} \leq n^{(k)}.$$

(51)

Next, we show that the dynamic regret of $\mathcal{A}^k$ with switching cost is almost optimal. From Theorem 7, we have

$$\sum_{t=r}^{s} \left( f_t(\mathbf{w}_t^k) + \lambda G \| \mathbf{w}_t^k - \mathbf{w}_{t+1}^k \| - f_t(\mathbf{u}_t) \right)$$

$$\leq \frac{D^2}{2\eta^{(k)}} + \frac{D}{\eta^{(k)}} P_{r,s} + \frac{(1 + 2\lambda)\eta^{(k)}\tau G^2}{2}$$

(52)

$$\overset{(29)}{=} \frac{G(D + 2P_{r,s})}{2}\sqrt{(1 + 2\lambda)n^{(k)}} + \frac{GD\tau}{2}\sqrt{\frac{1 + 2\lambda}{n^{(k)}}}$$

$$\overset{(51)}{\leq} \frac{(\sqrt{2} + 1)GD}{2}\sqrt{(1 + 2\lambda)\tau(1 + 2P_{r,s}/D)} \leq 2GD\sqrt{(1 + \lambda)\tau(1 + 2P_{r,s}/D)}.$$

Then, we prove that the regret of $\mathcal{B}^K$ w.r.t. $\mathcal{A}^k$ is roughly on the same order as (52). From Lemma 3, we have

$$\sum_{t=r}^{s}\left(f_t(\mathbf{v}_t^K)+\lambda G\|\mathbf{v}_t^K-\mathbf{v}_{t+1}^K\|\right)-\sum_{t=r}^{s}\left(f_t(\mathbf{w}_t^k)+\lambda G\|\mathbf{w}_t^k-\mathbf{w}_{t+1}^k\|\right)$$

$$\overset{(51),(45)}{\leq}GD\max(\sqrt{\lambda},1)\left((65+24P_{r,s}/D)\sqrt{n^{(k)}\log T}+15+12P_{r,s}/D\right)+$$

$$6GD\max(\sqrt{\lambda},1)(K-k)$$

$$\overset{(51)}{\leq}GD\max(\sqrt{\lambda},1)\left(65\sqrt{2\tau(1+2P_{r,s}/D)\log T}+15+12P_{r,s}/D\right) \tag{53}$$

$$+6GD\max(\sqrt{\lambda},1)(K-k)$$

$$\leq 114GD\max(\sqrt{\lambda},1)\sqrt{\tau(1+2P_{r,s}/D)\log T}+6GD\max(\sqrt{\lambda},1)\log_2\tau$$

$$\overset{\log_2\tau\leq\sqrt{\tau\log\tau}}{\leq}120GD\max(\sqrt{\lambda},1)\sqrt{\tau(1+2P_{r,s}/D)\log T}$$

where in the penultimate step we use the following inequalities

$$15+12P_{r,s}/D\overset{P_{r,s}\leq\tau D}{\leq}15+12\sqrt{\tau P_{r,s}/D}\overset{a+b\leq\sqrt{2a^2+2b^2}}{\leq}15\sqrt{2\tau(1+2P_{r,s}/D)},$$

$$K-k\overset{(32),(50)}{=}\left\lfloor\log_2\frac{T}{32\max(\lambda,1)\log 1/Z}\right\rfloor-\left\lfloor\log_2\frac{T(1+2P_{r,s}/D)}{\tau}\right\rfloor\leq\log_2\tau.$$

Combining (52) and (53), we can bound the dynamic regret of $\mathcal{B}^K$ with switching cost by

$$\sum_{t=r}^{s}\left(f_t(\mathbf{w}_t^K)+\lambda G\|\mathbf{w}_t^K-\mathbf{w}_{t+1}^K\|-f_t(\mathbf{u}_t)\right) \tag{54}$$

$$\leq 2GD\sqrt{(1+\lambda)\tau(1+2P_{r,s}/D)}+120GD\max(\sqrt{\lambda},1)\sqrt{\tau(1+2P_{r,s}/D)\log T}.$$

**Short Intervals** We consider short intervals $[r,s]$ such that

$$\frac{\tau}{1+2P_{r,s}/D}\geq 32\max(\lambda,1)\log\frac{1}{Z}.$$

Following the analysis of Theorem 3, the dynamic regret of $B^K$ over $[r,s]$ can be bounded by

$$\sum_{t=r}^{s}\left(f_t(\mathbf{w}_t^K)+\lambda G\|\mathbf{w}_t^K-\mathbf{w}_{t+1}^K\|-f_t(\mathbf{u}_t)\right)$$

$$\leq 3\tau GD\leq 3GD\sqrt{\tau\cdot 32\max(\lambda,1)\log T\cdot(1+2P_{r,s}/D)} \tag{55}$$

$$\leq 17GD\max(\sqrt{\lambda},1)\sqrt{\tau(1+2P_{r,s}/D)\log T}.$$

We complete the proof by combing (54) and (55).

### A.5 Proof of Theorem 6

From the standard analysis of OGD [Zinkevich, 2003], we have the following regret bound

$$\sum_{t=r}^{s}\left(f_t(\mathbf{w}_t)-f_t(\mathbf{w})\right)\leq\frac{D^2}{2\eta}+\frac{\eta(s-r+1)G^2}{2}. \tag{56}$$

To bound the switching cost, we have

$$\sum_{t=r}^{s}\|\mathbf{w}_t-\mathbf{w}_{t+1}\|=\sum_{t=r}^{s}\left\|\mathbf{w}_t-\Pi_\mathcal{W}\left[\mathbf{w}_t-\eta\nabla f_t(\mathbf{w}_t)\right]\right\|$$

$$\leq\sum_{t=r}^{s}\|-\eta\nabla f_t(\mathbf{w}_t)\|=\eta\sum_{t=r}^{s}\|\nabla f_t(\mathbf{w}_t)\|\overset{(19)}{\leq}\eta(s-r+1)G. \tag{57}$$

From (56) and (57), we have

$$\sum_{t=r}^{s} \big(f_t(\mathbf{w}_t) + \lambda G \|\mathbf{w}_t - \mathbf{w}_{t+1}\| - f_t(\mathbf{w})\big) \le \frac{D^2}{2\eta} + \frac{\eta(s-r+1)G^2}{2} + \lambda\eta(s-r+1)G^2.$$

## A.6 Proof of Theorem 7

From the dynamic regret of OGD [Zinkevich, 2003], in particular Theorem 6 of Zhang et al. [2018b], we have

$$\sum_{t=r}^{s} \big(f_t(\mathbf{w}_t) - f_t(\mathbf{u}_t)\big) \le \frac{D^2}{2\eta} + \frac{D}{\eta} \sum_{t=r}^{s} \|\mathbf{u}_t - \mathbf{u}_{t+1}\| + \frac{\eta(s-r+1)}{2}G^2.$$

We complete the proof by combining the above inequality with (57).

## A.7 Proof of Lemma 1

Similar to the analysis of Daniely and Mansour [2019, Theorem 22], we decompose the weighted sum of hitting cost and switching cost as

$$
\begin{aligned}
&f_t(\mathbf{w}_t) + \lambda G \|\mathbf{w}_t - \mathbf{w}_{t+1}\| \\
={}& f_t\big((1-w_t)\mathbf{w}_t^1 + w_t\mathbf{w}_t^2\big) + \lambda G \left\| (1-w_t)\mathbf{w}_t^1 + w_t\mathbf{w}_t^2 - (1-w_{t+1})\mathbf{w}_{t+1}^1 - w_{t+1}\mathbf{w}_{t+1}^2 \right\| \\
\le{}& (1-w_t)f_t(\mathbf{w}_t^1) + w_t f_t(\mathbf{w}_t^2) + \lambda G \left\| (1-w_t)(\mathbf{w}_t^1 - \mathbf{w}_{t+1}^1) \right\| + \lambda G \left\| w_t(\mathbf{w}_t^2 - \mathbf{w}_{t+1}^2) \right\| \\
&+ \lambda G \left\| (1-w_t)\mathbf{w}_{t+1}^1 - (1-w_{t+1})\mathbf{w}_{t+1}^1 + w_t\mathbf{w}_{t+1}^2 - w_{t+1}\mathbf{w}_{t+1}^2 \right\| \\
={}& (1-w_t)\big(f_t(\mathbf{w}_t^1) + \lambda G \|\mathbf{w}_t^1 - \mathbf{w}_{t+1}^1\|\big) + w_t\big(f_t(\mathbf{w}_t^2) + \lambda G \|\mathbf{w}_t^2 - \mathbf{w}_{t+1}^2\|\big) \\
&+ \lambda G \left\| (w_t - w_{t+1})(\mathbf{w}_{t+1}^1 - \mathbf{w}_{t+1}^2) \right\| \\
\overset{(20)}{\le}{}& (1-w_t)\big(f_t(\mathbf{w}_t^1) + \lambda G \|\mathbf{w}_t^1 - \mathbf{w}_{t+1}^1\|\big) + w_t\big(f_t(\mathbf{w}_t^2) + \lambda G \|\mathbf{w}_t^2 - \mathbf{w}_{t+1}^2\|\big) \\
&+ \lambda GD|w_t - w_{t+1}|.
\end{aligned}
\tag{58}
$$

Then, the regret of $\mathcal{A}$ w.r.t. $\mathcal{A}^1$ over any interval $[r, s]$ can be upper bounded in the following way:

$$
\begin{aligned}
&\sum_{t=r}^{s} \big(f_t(\mathbf{w}_t) + \lambda G \|\mathbf{w}_t - \mathbf{w}_{t+1}\|\big) - \sum_{t=r}^{s} \big(f_t(\mathbf{w}_t^1) + \lambda G \|\mathbf{w}_t^1 - \mathbf{w}_{t+1}^1\|\big) \\
\overset{(58)}{\le}{}& \sum_{t=r}^{s} \Big( w_t \big[ (f_t(\mathbf{w}_t^2) + \lambda G \|\mathbf{w}_t^2 - \mathbf{w}_{t+1}^2\|) - (f_t(\mathbf{w}_t^1) + \lambda G \|\mathbf{w}_t^1 - \mathbf{w}_{t+1}^1\|) \big] \\
&+ \lambda GD|w_t - w_{t+1}| \Big) \\
\overset{(26),(27)}{=}{}& \sum_{t=r}^{s} \big( w_t(\ell_t^2 - \ell_t^1) + \lambda GD|w_t - w_{t+1}| \big) \\
\overset{(28)}{=}{}& -(1+M)GD \sum_{t=r}^{s} \Big( w_t\ell_t - \frac{\lambda}{1+M}|w_t - w_{t+1}| \Big) \\
\le{}& -(1+M)GD \sum_{t=r}^{s} \big( w_t\ell_t - \lambda|w_t - w_{t+1}| \big)
\end{aligned}
$$

which proves (24). Similarly, the regret of $\mathcal{A}$ w.r.t. $\mathcal{A}^2$ over any interval $[r, s]$ can be upper bounded by

$$\sum_{t=r}^{s} \left( f_t(\mathbf{w}_t) + \lambda G \|\mathbf{w}_t - \mathbf{w}_{t+1}\| \right) - \sum_{t=r}^{s} \left( f_t(\mathbf{w}_t^2) + \lambda G \|\mathbf{w}_t^2 - \mathbf{w}_{t+1}^2\| \right)$$

$$\stackrel{(58)}{\leq} \sum_{t=r}^{s} (1 - w_t) \left[ \left( f_t(\mathbf{w}_t^1) + \lambda G \|\mathbf{w}_t^1 - \mathbf{w}_{t+1}^1\| \right) - \left( f_t(\mathbf{w}_t^2) + \lambda G \|\mathbf{w}_t^2 - \mathbf{w}_{t+1}^2\| \right) \right]$$

$$+ \sum_{t=r}^{s} \lambda G D |w_t - w_{t+1}|$$

$$\stackrel{(26),(27)}{=} \sum_{t=r}^{s} \left( (1 - w_t)(\ell_t^1 - \ell_t^2) + \lambda G D |w_t - w_{t+1}| \right)$$

$$\stackrel{(28)}{=} -(1 + M) G D \sum_{t=r}^{s} \left( w_t \ell_t - \frac{\lambda}{1 + M} |w_t - w_{t+1}| - \ell_t \right)$$

$$\leq -(1 + M) G D \sum_{t=r}^{s} \left( w_t \ell_t - \lambda |w_t - w_{t+1}| - \ell_t \right)$$

which proves (25).

### A.8 Proof of Lemma 2

We will prove that the outputs of $\mathcal{A}^i$'s and $\mathcal{B}^i$'s move slowly such that (23) holds. Let $\mathbf{w}_t^i$ be the output of $\mathcal{A}^i$ in the $t$-th iteration. From the updating rule of OGD, we have

$$\|\mathbf{w}_t^i - \mathbf{w}_{t+1}^i\| \leq \eta^{(i)} \left\| \nabla f_t(\mathbf{w}_t^i) \right\| \stackrel{(19)}{\leq} \eta^{(i)} G \stackrel{(29)}{=} D \sqrt{\frac{1}{(1 + 2\lambda) n^{(i)}}} \stackrel{(41)}{\leq} \frac{D}{\lambda}. \tag{59}$$

So, $\mathbf{w}_t^i$'s satisfy the condition in (23) when $M = 2$.

Let $\mathbf{v}_t^i$ be the output of $\mathcal{B}^i$ in the $t$-th iteration. We will prove by induction that

$$\|\mathbf{v}_t^i - \mathbf{v}_{t+1}^i\| \leq \frac{D}{\lambda} + \frac{D}{\max(\sqrt{\lambda}, 1)} \sum_{j=2}^{i} \left( \sqrt{\frac{1}{n^{(j)}} \log \frac{1}{Z}} + \frac{Z}{4} \right), \quad \forall i \in [K]. \tag{60}$$

The above equation, together with the following fact

$$\frac{D}{\lambda} + \frac{D}{\max(\sqrt{\lambda}, 1)} \sum_{j=2}^{K} \left( \sqrt{\frac{1}{n^{(j)}} \log \frac{1}{Z}} + \frac{Z}{4} \right)$$

$$\stackrel{(30)}{=} \frac{D}{\lambda} + \frac{D}{\max(\sqrt{\lambda}, 1)} \sqrt{\frac{1}{2T} \log \frac{1}{Z}} \sum_{j=2}^{K} \sqrt{2^j} + \frac{D}{\max(\sqrt{\lambda}, 1)} \frac{Z(K-1)}{4}$$

$$\leq \frac{D}{\lambda} + \frac{D}{\max(\sqrt{\lambda}, 1)} \sqrt{\frac{1}{2T} \log \frac{1}{Z}} \frac{2}{\sqrt{2} - 1} \sqrt{2}^{K-1} + \frac{D}{\max(\sqrt{\lambda}, 1)} \frac{Z(K-1)}{4} \tag{61}$$

$$\stackrel{(32)}{\leq} \frac{D}{\lambda} + \frac{D}{\max(\sqrt{\lambda}, 1)} \sqrt{\frac{1}{2T} \log \frac{1}{Z}} \frac{2}{\sqrt{2} - 1} \sqrt{\frac{T}{32\lambda \log 1/Z}} + \frac{D}{\max(\sqrt{\lambda}, 1)} \frac{Z}{4} \log_2 T$$

$$= \frac{D}{\lambda} + \frac{(\sqrt{2} + 1)D}{4 \max(\sqrt{\lambda}, 1)\sqrt{\lambda}} + \frac{D \log_2 T}{4 \max(\sqrt{\lambda}, 1)T} \stackrel{(31)}{\leq} \frac{2D}{\lambda}$$

implies that $\mathbf{v}_t^i$'s meet the condition in (23) when $M = 2$.

Since $\mathcal{B}^1 = A^1$, we have

$$\|\mathbf{v}_t^1 - \mathbf{v}_{t+1}^1\| = \|\mathbf{w}_t^1 - \mathbf{w}_{t+1}^1\| \stackrel{(59)}{\leq} \frac{D}{\lambda}. \tag{62}$$

Thus, (60) holds when $i = 1$. Suppose (60) is true when $i = k$, and thus

$$\|\mathbf{v}_t^k - \mathbf{v}_{t+1}^k\| \leq \frac{D}{\lambda} + \frac{D}{\max(\sqrt{\lambda}, 1)} \sum_{j=2}^k \left( \sqrt{\frac{1}{n^{(j)}} \log \frac{1}{Z}} + \frac{Z}{4} \right) \overset{(61)}{\leq} \frac{2D}{\lambda}. \tag{63}$$

We proceed to bound the movement of $\mathbf{v}_t^{k+1}$, which is the output of $B^{k+1}$. Recall that $B^{k+1}$ is an instance of Combiner which aggregates $\mathcal{B}^k$ and $\mathcal{A}^{k+1}$. From the procedure of Algorithm 3, we have

$$\mathbf{v}_t^{k+1} \overset{(22)}{=} (1 - w_t^{k+1})\mathbf{v}_t^k + w_t^{k+1}\mathbf{w}_t^{k+1}$$

where $w_t^{k+1}$ is the weight generated by DNP-cu. Thus, the movement of $\mathbf{v}_t^{k+1}$ can be bounded by

$$\begin{aligned}
\|\mathbf{v}_t^{k+1} - \mathbf{v}_{t+1}^{k+1}\| &= \left\| (1 - w_t^{k+1})\mathbf{v}_t^k + w_t^{k+1}\mathbf{w}_t^{k+1} - \left( (1 - w_{t+1}^{k+1})\mathbf{v}_{t+1}^k + w_{t+1}^{k+1}\mathbf{w}_{t+1}^{k+1} \right) \right\| \\
&\leq \left\| (1 - w_t^{k+1})\mathbf{v}_t^k - (1 - w_{t+1}^{k+1})\mathbf{v}_t^k + w_t^{k+1}\mathbf{w}_t^{k+1} - w_{t+1}^{k+1}\mathbf{w}_t^{k+1} \right\| \\
&\quad + \left\| (1 - w_{t+1}^{k+1})\mathbf{v}_t^k + w_{t+1}^{k+1}\mathbf{w}_t^{k+1} - \left( (1 - w_{t+1}^{k+1})\mathbf{v}_{t+1}^k + w_{t+1}^{k+1}\mathbf{w}_{t+1}^{k+1} \right) \right\| \\
&\leq |w_t^{k+1} - w_{t+1}^{k+1}| \|\mathbf{v}_t^k - \mathbf{w}_t^{k+1}\| + (1 - w_{t+1}^{k+1})\|\mathbf{v}_t^k - \mathbf{v}_{t+1}^k\| + w_{t+1}^{k+1}\|\mathbf{w}_t^{k+1} - \mathbf{w}_{t+1}^{k+1}\| \\
&\overset{(20),(59)}{\leq} D|w_t^{k+1} - w_{t+1}^{k+1}| + (1 - w_{t+1}^{k+1})\|\mathbf{v}_t^k - \mathbf{v}_{t+1}^k\| + w_{t+1}^{k+1}\frac{D}{\lambda} \\
&\leq D|w_t^{k+1} - w_{t+1}^{k+1}| + \max\left( \|\mathbf{v}_t^k - \mathbf{v}_{t+1}^k\|, \frac{D}{\lambda} \right).
\end{aligned} \tag{64}$$

From (59) and (63), we know that the outputs of $\mathcal{B}^k$ and $\mathcal{A}^{k+1}$ satisfy (23). Thus, we can apply Corollary 2 to bound the change of $w_t^{k+1}$:

$$|w_t^{k+1} - w_{t+1}^{k+1}| \overset{(16)}{\leq} \frac{1}{\max(\sqrt{\lambda}, 1)} \left( \sqrt{\frac{1}{n^{(k+1)}} \log \frac{1}{Z}} + \frac{Z}{4} \right). \tag{65}$$

From (64) and (65), we have

$$\begin{aligned}
\|\mathbf{v}_t^{k+1} - \mathbf{v}_{t+1}^{k+1}\| &\leq \frac{D}{\max(\sqrt{\lambda}, 1)} \left( \sqrt{\frac{1}{n^{(k+1)}} \log \frac{1}{Z}} + \frac{Z}{4} \right) + \max\left( \|\mathbf{v}_t^k - \mathbf{v}_{t+1}^k\|, \frac{D}{\lambda} \right) \\
&\overset{(63)}{\leq} \frac{D}{\lambda} + \frac{D}{\max(\sqrt{\lambda}, 1)} \sum_{j=2}^{k+1} \left( \sqrt{\frac{1}{n^{(j)}} \log \frac{1}{Z}} + \frac{Z}{4} \right)
\end{aligned}$$

which shows that (60) holds when $i = k + 1$.

## A.9 Proof of Lemma 3

The regret of $\mathcal{B}^K$ w.r.t. $\mathcal{A}^k$ can be decomposed as

$$\begin{aligned}
&\sum_{t=r}^s \left( f_t(\mathbf{v}_t^K) + \lambda G \|\mathbf{v}_t^K - \mathbf{v}_{t+1}^K\| \right) - \sum_{t=r}^s \left( f_t(\mathbf{w}_t^k) + \lambda G \|\mathbf{w}_t^k - \mathbf{w}_{t+1}^k\| \right) \\
&= \underbrace{\sum_{t=r}^s \left( f_t(\mathbf{v}_t^k) + \lambda G \|\mathbf{v}_t^k - \mathbf{v}_{t+1}^k\| \right) - \sum_{t=r}^s \left( f_t(\mathbf{w}_t^k) + \lambda G \|\mathbf{w}_t^k - \mathbf{w}_{t+1}^k\| \right)}_{:=U} \\
&\quad + \sum_{i=k+1}^K \left( \underbrace{\sum_{t=r}^s \left( f_t(\mathbf{v}_t^i) + \lambda G \|\mathbf{v}_t^i - \mathbf{v}_t^{i+1}\| \right) - \sum_{t=r}^s \left( f_t(\mathbf{v}_t^{i-1}) + \lambda G \|\mathbf{v}_t^{i-1} - \mathbf{v}_{t+1}^{i-1}\| \right)}_{:=V^i} \right)
\end{aligned} \tag{66}$$

where $U$ is the regret of $\mathcal{B}^k$ w.r.t. $\mathcal{A}^k$, and $V^i$ is the regret of $\mathcal{B}^i$ w.r.t. $\mathcal{B}^{i-1}$. Next, we make use of Corollary 2 and Lemma 1 to bound these quantities.

To bound $U$, we have

$$U \overset{(15),(25)}{\leq} 3GD\left(\max(\sqrt{\lambda},1)U(n^{(k)})\left(\frac{\tau}{n^{(k)}}+1\right)+\frac{2\tau}{n^{(k)}}+1+\max(\sqrt{\lambda},1)U(n^{(k)})\right.$$
$$\left.+1+\max(\sqrt{\lambda},1)\frac{\tau}{T}\right)$$
$$\overset{\tau\leq cn^{(k)}}{\leq} 3GD\left((2+c)\max(\sqrt{\lambda},1)U(n^{(k)})+2+2c+\max(\sqrt{\lambda},1)\right) \tag{67}$$
$$\overset{(10)}{\leq} 3GD\left((2+c)\max(\sqrt{\lambda},1)\sqrt{16n^{(k)}\log T}+2+2c+\max(\sqrt{\lambda},1)\right)$$
$$\leq 3GD\max(\sqrt{\lambda},1)\left(4(2+c)\sqrt{n^{(k)}\log T}+3+2c\right).$$

To bound the summation of $V^i$, we have

$$\sum_{i=k+1}^{K}V^i \overset{(15),(24)}{\leq} 3GD\sum_{i=k+1}^{K}\left(\max(\sqrt{\lambda},1)U(n^{(i)})+1+\max(\sqrt{\lambda},1)\frac{\tau}{T}\right)$$
$$\leq 3GD\max(\sqrt{\lambda},1)\sum_{i=k+1}^{K}U(n^{(i)})+6GD\max(\sqrt{\lambda},1)(K-k)$$
$$\overset{(10)}{\leq} 3GD\max(\sqrt{\lambda},1)\sum_{i=k+1}^{K}\sqrt{16n^{(i)}\log T}+6GD\max(\sqrt{\lambda},1)(K-k) \tag{68}$$
$$\overset{(30)}{=} 12GD\max(\sqrt{\lambda},1)\sqrt{n^{(k)}\log T}\sum_{i=1}^{K-k}\sqrt{2^{-i}}+6GD\max(\sqrt{\lambda},1)(K-k)$$
$$\leq 12GD\max(\sqrt{\lambda},1)\sqrt{n^{(k)}\log T}\frac{1}{\sqrt{2}-1}+6GD\max(\sqrt{\lambda},1)(K-k)$$
$$\leq 29GD\max(\sqrt{\lambda},1)\sqrt{n^{(k)}\log T}+6GD\max(\sqrt{\lambda},1)(K-k).$$

We complete the proof by substituting (67) and (68) into (66).

## A.10 Proof of Theorem 5

Our purpose is to provide a general analysis of Algorithm 1 over any bit sequence, so we do not make use of the range of $x_t$ in (33). As an alternative, we use the following simple upper bound

$$|x_t|\leq\mu n,\ \forall t\geq 1 \tag{69}$$

which can be proved by induction. From the initialization, we have $|x_1|=0\leq\mu n$. Now, suppose $|x_k|\leq\mu n$. Then, we have

$$|x_{k+1}|\leq|\rho x_k|+|b_k| \overset{\rho=1-1/n}{\leq}\left(1-\frac{1}{n}\right)\mu n+\mu=\mu n.$$

Then, we can bound the difference between any two consecutive derivations by 2:

$$|x_t-x_{t+1}|=|(1-\rho)x_t-b_t|\leq\frac{1}{n}|x_t|+|b_t| \overset{(69)}{\leq} 2\mu\leq 2. \tag{70}$$

Next, we introduce Lemma 19 of Daniely and Mansour [2019], which characterizes the derivative of $g(\cdot)$ over short intervals.

**Lemma 4** *Suppose* $\log\frac{1}{Z}\leq\frac{n}{16}$, $Z\leq\frac{1}{e}$ *and* $n\geq 8e$. *For every segment* $\mathcal{I}\subset\mathbb{R}$ *of length* $\leq 2$ *and every* $x\in\mathcal{I}$, *we have*

$$4\max_{s\in\mathcal{I}}|g'(s)|\leq\frac{1}{n}xg(x)+Z.$$

Then, we can apply the above lemma to bound the derivative of $g(\cdot)$ over the interval $[x_t, x_{t+1}],$[4] whose length is smaller than 2. Under the conditions of Lemma 4, we have

$$4 \max_{s \in [x_t, x_{t+1}]} |g'(s)| \leq \frac{1}{n} x_t g(x_t) + Z. \tag{71}$$

Since $g(x) = 0$ if $x \leq 0$, we have

$$4 \max_{s \in [x_t, x_{t+1}]} |g'(s)| \leq Z, \text{ if } x_t \leq 0. \tag{72}$$

Furthermore, we know that $g'(x) = 0$, if $x \geq U(n)$. When $x_t \geq U(n) + 2\mu$, from (70) we have

$$[x_t, x_{t+1}] \subset [U(n), \infty).$$

Thus,

$$\max_{s \in [x_t, x_{t+1}]} |g'(s)| = 0, \text{ if } x_t \geq U(n) + 2\mu. \tag{73}$$

Let $\mathbb{I}(x)$ be the indicator function of the interval $[0, U(n) + 2\mu]$. We can summarize the general result in (71) and the special cases in (72) and (73) as

$$4 \max_{s \in [x_t, x_{t+1}]} |g'(s)| \leq \frac{1}{n} x_t g(x_t) \mathbb{I}(x_t) + Z. \tag{74}$$

We proceed to use the following potential function

$$\Phi_t = \int_0^{x_t} g(s) ds$$

to analyze the reward of Algorithm 1. It is easy to verify that

$$\max(0, x_t - U(n)) \leq \Phi_t = \int_0^{x_t} g(s) ds \leq \max(x_t, 0). \tag{75}$$

To bound the change of the potential function, we need the following inequality for piece-wise differential functions $f : [a, b] \mapsto \mathbb{R}$ [Kapralov and Panigrahy, 2010, Daniely and Mansour, 2019]

$$\int_a^b f(x) dx \leq f(a)(b - a) + \max |f'(z)| \frac{1}{2} (b - a)^2. \tag{76}$$

We have

$$
\begin{aligned}
\Phi_{t+1} - \Phi_t &= \int_{x_t}^{x_{t+1}} g(s) ds \\
&\stackrel{(76)}{\leq} g(x_t)(x_{t+1} - x_t) + \frac{1}{2}(x_{t+1} - x_t)^2 \max_{s \in [x_t, x_{t+1}]} |g'(s)| \\
&\stackrel{(70)}{\leq} g(x_t)\left(-\frac{1}{n} x_t + b_t\right) + 2 \max_{s \in [x_t, x_{t+1}]} |g'(s)| \\
&= g(x_t)\left(-\frac{1}{n} x_t + b_t\right) - 2 \max_{s \in [x_t, x_{t+1}]} |g'(s)| + 4 \max_{s \in [x_t, x_{t+1}]} |g'(s)| \\
&\leq g(x_t)\left(-\frac{1}{n} x_t + b_t\right) - 2 \left| \frac{g(x_t) - g(x_{t+1})}{x_t - x_{t+1}} \right| + 4 \max_{s \in [x_t, x_{t+1}]} |g'(s)| \\
&\stackrel{(70)}{\leq} g(x_t)\left(-\frac{1}{n} x_t + b_t\right) - \frac{1}{\mu} |g(x_t) - g(x_{t+1})| + 4 \max_{s \in [x_t, x_{t+1}]} |g'(s)| \\
&\stackrel{(74)}{\leq} g(x_t)\left(-\frac{1}{n} x_t + b_t\right) - \frac{1}{\mu} |g(x_t) - g(x_{t+1})| + \frac{1}{n} x_t g(x_t) \mathbb{I}(x_t) + Z \\
&= g(x_t) b_t - \frac{1}{\mu} |g(x_t) - g(x_{t+1})| + \frac{1}{n} x_t g(x_t) \left(\mathbb{I}(x_t) - 1\right) + Z
\end{aligned}
\tag{77}
$$

---

[4]With a slight abuse of notation, we will write $[a, b]$ to denote $[\min\{a, b\}, \max\{a, b\}]$.

where the 3rd inequality is due to the mean value theorem. To bound the cumulative reward over any interval $[r, s]$, we sum (77) from $t = r$ to $t = s$, and obtain

$$\Phi_{s+1} - \Phi_r \leq \sum_{t=r}^{s} \left( g(x_t)b_t - \frac{1}{\mu}|g(x_t) - g(x_{t+1})| \right) + \sum_{t=r}^{s} \frac{1}{n}x_t g(x_t)\left(\mathbb{I}(x_t) - 1\right) + Z\tau.$$

Thus,

$$\sum_{t=r}^{s} \left( g(x_t)b_t - \frac{1}{\mu}|g(x_t) - g(x_{t+1})| \right) \geq \Phi_{s+1} + \sum_{t=r}^{s} \frac{1}{n}x_t g(x_t)\left(1 - \mathbb{I}(x_t)\right) - \Phi_r - Z\tau. \quad (78)$$

First, we use the simple fact that

$$x_t g(x_t)\left(1 - \mathbb{I}(x_t)\right) \geq 0,$$

to simplify (78), and have

$$\sum_{t=r}^{s} \left( g(x_t)b_t - \frac{1}{\mu}|g(x_t) - g(x_{t+1})| \right) \geq -\Phi_r - Z\tau. \quad (79)$$

Second, we lower bound the cumulative reward by the summation of the bit sequence. From (75), we have

$$\Phi_{s+1} \geq x_{s+1} - U(n). \quad (80)$$

We also have

$$x_t g(x_t)\left(1 - \mathbb{I}(x_t)\right) \geq x_t - U(n) - 2\mu. \quad (81)$$

That is because if $x_t \geq U(n) + 2\mu$, we have

$$x_t g(x_t)\left(1 - \mathbb{I}(x_t)\right) = x_t \geq x_t - U(n) - 2\mu;$$

otherwise,

$$x_t g(x_t)\left(1 - \mathbb{I}(x_t)\right) \geq 0 \geq x_t - U(n) - 2\mu.$$

Based on (80) and (81), we have

$$\Phi_{s+1} + \sum_{t=r}^{s} \frac{1}{n}x_t g(x_t)\left(1 - \mathbb{I}(x_t)\right)$$

$$\geq x_{s+1} - U(n) + \frac{1}{n}\sum_{t=r}^{s}(x_t - U(n) - 2\mu) = x_{s+1} + \frac{1}{n}\sum_{t=r}^{s} x_t - \frac{\tau}{n}\left(U(n) + 2\mu\right) - U(n)$$

$$= \rho^\tau x_r + \sum_{j=r}^{s}\rho^{s-j}b_j + \frac{1}{n}\sum_{t=r}^{s}\left(\rho^{t-r}x_r + \sum_{j=r}^{t-1}\rho^{t-1-j}b_j\right) - \frac{\tau}{n}\left(U(n) + 2\mu\right) - U(n)$$

$$= \rho^\tau x_r + \sum_{j=r}^{s}\rho^{s-j}b_j + \frac{x_r}{n}\sum_{t=r}^{s}\rho^{t-r} + \sum_{j=r}^{s}\frac{b_j}{n}\sum_{t=j+1}^{s}\rho^{t-1-j} - \frac{\tau}{n}\left(U(n) + 2\mu\right) - U(n)$$

$$= \rho^\tau x_r + \sum_{j=r}^{s}\rho^{s-j}b_j + \frac{x_r}{n}\frac{1 - \rho^\tau}{1 - \rho} + \sum_{j=r}^{s}\frac{b_j}{n}\frac{1 - \rho^{s-j}}{1 - \rho} - \frac{\tau}{n}\left(U(n) + 2\mu\right) - U(n)$$

$$\stackrel{\rho=1-1/n}{=} x_r + \sum_{t=r}^{s}b_t - \frac{\tau}{n}\left(U(n) + 2\mu\right) - U(n).$$

Combining the above inequality with (78), we have

$$\sum_{t=r}^{s} \left( g(x_t)b_t - \frac{1}{\mu}|g(x_t) - g(x_{t+1})| \right) \geq \sum_{t=r}^{s} b_t + x_r - \frac{\tau}{n}\left(U(n) + 2\mu\right) - U(n) - \Phi_r - Z\tau.$$

$$(82)$$

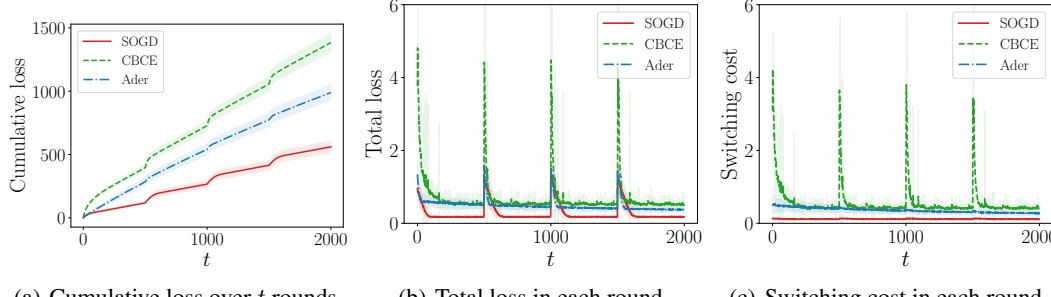

| (a) Cumulative loss over $t$ rounds | (b) Total loss in each round | (c) Switching cost in each round |

Figure 1: Performance of different methods versus the number of rounds.

Third, from (79) and (82), we have

$$\sum_{t=r}^{s} \left( g(x_t) b_t - \frac{1}{\mu} |g(x_t) - g(x_{t+1})| \right)$$

$$\geq \max \left( 0, \sum_{t=r}^{s} b_t + x_r - \frac{\tau}{n} \left( U(n) + 2\mu \right) - U(n) \right) - \Phi_r - Z\tau$$

$$\overset{(75)}{\geq} \max \left( 0, \sum_{t=r}^{s} b_t + x_r - \frac{\tau}{n} \left( U(n) + 2\mu \right) - U(n) \right) - \max(x_r, 0) - Z\tau$$

which proves (34).

Finally, to bound the change of successive predictions, we have

$$|g(x_t) - g(x_{t+1})| \leq |x_t - x_{t+1}| \max_s |g'(s)| \overset{(70)}{\leq} 2\mu \max_s |g'(s)|. \tag{83}$$

Following the analysis of Lemma 23 of Daniely and Mansour [2019], we know that $g'(\cdot)$ is nondecreasing in $[0, U(n)]$ and is 0 outside, and thus

$$\max_s |g'(s)| = g'(U(n)) = \frac{U(n)g(U(n))}{8n} + \frac{Z}{8} = \frac{U(n)}{8n} + \frac{Z}{8} \overset{(10)}{\leq} \frac{\sqrt{16n \log \frac{1}{Z}}}{8n} + \frac{Z}{8} \tag{84}$$

where the 2nd equality is due to the property of the confidence function [Daniely and Mansour, 2019, Lemma 18]. We obtain (35) by combining (83) and (84).

## B Experiments

In this section, we implement online linear regression on synthetic data to evaluate our method, i.e., smoothed OGD (SOGD). In each round $t$, a batch of data points $\{(\mathbf{x}_{t,1}, y_{t,1}), \ldots, (\mathbf{x}_{t,n}, y_{t,n})\}$ arrive, where $\mathbf{x}_{t,i}$ is sampled randomly from $[-1, 1]^d$. The target value $y_{t,i}$ is generated by $y_{t,i} = \mathbf{w}^\top \mathbf{x}_{t,i} + \epsilon$, where $\epsilon \sim \mathcal{N}(0, 0.1)$ is a zero-mean Gaussian noise with standard deviation 0.1. The unknown parameter $\mathbf{w}$ is sampled randomly from $[-1, 1]^d$, and would be re-sampled every 500 rounds to simulate changing environments. After predicting $\mathbf{w}_t$, the online learner suffers the following total loss

$$\sum_{i=1}^{n} |\mathbf{w}_t^\top \mathbf{x}_{t,i} - y_{t,i}| + \lambda G \|\mathbf{w}_t - \mathbf{w}_{t-1}\|$$

which includes both the hitting cost and the switching cost.

In the experiment, we set $n = 64$, $d = 10$, $\lambda = 1$, $D = 2\sqrt{10}$, and $G = \sqrt{10}$. We compare our method with CBCE [Jun et al., 2017a] and Ader [Zhang et al., 2018a], which obtain optimal adaptive regret and dynamic regret respectively, but do not consider the switching cost.

The whole experiment is conducted on a personal laptop equipped with an Intel i7-10750H CPU and 16G memory. We repeat the experiment $100$ times and plot the average cumulative loss, total loss and switching cost in Fig. 1. As can be seen, all three methods can deal with changing environments and adapt quickly when the underlying parameter $\mathbf{w}$ changes. Among them, our SOGD suffers the smallest cumulative loss, and incurs the least total loss in most rounds. As indicated in Fig. 1(c), both Ader and CBCE have much higher switching cost compared with our method, and CBCE suffers huge switching loss when $\mathbf{w}$ is re-sampled. In contrast, SOGD maintains the lowest switching cost in all rounds, since it explicitly takes the switching cost into consideration.