# OpenReview forum: "Smoothed Online Convex Optimization Based on Discounted-Normal-Predictor"
_NeurIPS.cc/2022/Conference — NeurIPS 2022 Accept_

### Official Review · Reviewer_m3LC · 2022-07-11

**Rating:** 7
**Confidence:** 3
**Soundness:** 3 good
**Presentation:** 3 good
**Contribution:** 3 good

**Summary:**

In this paper, the authors study algorithms for smoothed online convex optimization (SOCO). Compared to classical online convex optimization (OCO), in SOCO the player also suffer a switching cost between rounds depending on how much the iterates changed from one round to another. In this work the authors propose an algorithm the incurs low dynamic and adaptive regret simultaneously. To do so, they instantiate around $O(\log T)$ instances of online gradient descent with different step-sizes, and combine these algorithms using a technique called "Discounted-Normal-Predictor" originally developed for online bit prediction. Their results either match previous guarantees (in the case of adaptive regret), or are the one of the first of its kind for SOCO (for dynamic regret), matching the known lower-bounds up to logarithmic factors.

**Questions:**

- Could you help me understand which works on bit prediction were with bits on $[-1,1]$ and which ones were on $[0,1]$;
- Could the authors try to give intuition behind (10)? The presence of $U(n)$ and $\mu$ make it a bit harder to digest (and maybe with more time to read the paper I'd be able to get the intuition);

**Limitations:**

The authors do a good discussion, afaik, of related work and how their work can be placed on the literature. Maybe some discussion of computational efficiency/cost of the algorithm would have been interesting, but it certainly is not crucial.

**Strengths And Weaknesses:**

### Strengths
- The technique uses in a sleek method from online bit prediction to combine different OCO algorithms. Although this bit prediction method is a bit mysterious, the high-level idea of the final algorithm is well-explained and intuitive;
- The fact that one single algorithm manages to control both dynamic and adaptive regret simultaneously is neat;
- The paper is quite well-written, with an excellent literature review

### Weaknesses
- I noticed the authors did try to explain the main ideas behind the bit prediction strategy, but I had a very hard time parsing the bounds in Theorem 1. This influences later on the understanding of the remainder of the algorithm and the intuition behind some of its construction;
- I was a bit confused about when the $b_t$'s should be in $[0,1]$ and when they should be in $[-1,1]$. The beginning of sec 2.3 talks about it being in $[-1,1]$. But when describing the work of Daniely and Mansour, the predictions (coming from eq (7) according to Alg 1) end up in $[0,1]$. At the beginning of sec 3.1 the authors also say "(...) but require the prediction to lie in [0,1] (...)", what made me believe previous work was mostly on $[-1,1];
- Related to the confusions above, the remark on line 222 was not quite immediate.

### Minor observations
- In (9) you use $\hat{g}$ but never defined it. Was it supposed to be (7)?
- Line 165: what do you mean that the "property" of $g$ has been "revealed"?
- Line 216: you use "meta-regret" without defining it, which caused a bit of confusion when I was reading. Maybe something that sounds less formal like "the difference in the costs between the sub-algorithms and the main one" would avoid this;
- In Lemma 1 $\mathcal{A}$ is the final algorithm and in Algorithm 3 it is the bit prediction algorithm, which is confusing since Lemma 1 is about Algorithm 3;
- Using $\cap$ to denote "AND" was a bit unusual for me. If other reviewers find it weird as well, consider changing, but it may be just me in this case;
- It is a bit "unsettling" to call a real number a bit;

---

> ### Author Response · Authors · 2022-08-01
> **Many thanks for the constructive reviews!**
>
> Many thanks for the constructive reviews! We will revise our paper accordingly.
>
> ---
>
> Q1: Could you help me understand which works on bit prediction were with bits on $[-1,1]$ and which ones were on $[0,1]$;
>
> A1: Actually, there is no fundamental difference between the two cases. When introducing the bit prediction problem, Kapralov and Panigrahy [2010] assume that the bit belongs to $[-1,1]$, and later they discuss the case of $[0,1]$ in Section 4 of their paper. Daniely and Mansour [2019] focus on the $[0,1]$ case, but their algorithm can be extended to the $[-1,1]$ case by replacing the projection operation $\Pi_{[0,1]}$ in (7) with $\Pi_{[-1,1]}$.
>
> ---
>
> Q2: Could the authors try to give intuition behind (10)? The presence of $U(n)$ and $\mu$ make it a bit harder to digest;
>
> A2: Notice that (13) is almost the same as (10), expect for a scaling factor. So, to avoid duplication, we only discussed the implication of (13) in the **Remark** after Corollary 2. \
> Because there is a max operator in (13), we can derive two different lower bounds in (15) and (16), which can be used by Lemma 1 to bound the meta-regret. Based on (15) and (22), we can upper bound the regret of the meta-algorithm $\mathcal{A}$ w.r.t. the first algorithm $\mathcal{A}_1$. Similarly, from (16) and (23), we can upper bound the regret of $\mathcal{A}$ w.r.t. $\mathcal{A}_2$. When applying DNP-cu (i.e., Algorithm 3) to combine multiple algorithms sequentially, (15) and (16) play two different roles, as explained below.
> 1. Based on (15), we do not destroy the theoretical guarantee of early algorithms. Th reason is because the bound in (15) is *independent* of the interval length $\tau$, which is a very strong property. For details, please refer to (66) of the supplementary.
> 2. Based on (16), we can inherit the theoretical guarantee of the current algorithm. This is rather straightforward, and can be found in (65) of the supplementary.
>
> ---
>
> Q3: Related to the confusions above, the remark on line 222 was not quite immediate.
>
> A3: To facilitate understanding, we will provide the mappings below:
> $$
> w_t \mapsto g(x_t), \quad  \ell_t \mapsto b_t.
> $$
>
> ---
>
> Q4: In (9) you use $ \tilde{g}$ but never defined it. Was it supposed to be (7)?
>
> A4: We are sorry for the confusion. We accidentally delete the definition of $ \tilde{g}$, which is the function inside the $[\cdot]$ of (7), i.e.,
> $$
> \tilde{g}(x)= \sqrt{\frac{n}{8}} Z \cdot \textrm{erf} \left( \frac{x}{\sqrt{8 n }} \right) e^{\frac{x^2}{16 n}}
> $$
>
> ---
>
> Q5: Line 165: what do you mean that the "property" of $g$ has been "revealed"?
>
> A5: We mean that Daniely and Mansour [2019] have proved two lemmas (Lemma 18 and Lemma 19 in their paper) to demonstrate the property of $g$.
>
> ---
>
> Q6: In Lemma 1 $\mathcal{A}$ is the final algorithm and in Algorithm 3 it is the bit prediction algorithm, which is confusing since Lemma 1 is about Algorithm 3;
>
> A6: Thanks for the suggestion. We will change the notation.
>
> ---
>
> Q7: Using $\cap$ to denote "AND" was a bit unusual for me.
>
> A7: We would like to use & to denote “AND”.

---

> > ### Comment · Reviewer_m3LC · 2022-08-05
> > **Great reply and no further questions**
> >
> > I'd like to thank the authors for their thorough replies (not only to me but to the other reviewers, it does help with understanding the paper in spite of the short review period).
> >
> > Regarding the answer to Q1, this only makes me believe you should be a bit more careful when considering "bits" in $[0,1]$ and in $[-1,1]$. Since they do not have any significant impact, what happens when flipping between the two in the discussions in the paper is to needlessly confuse readers not deeply familiar with previous work. This is not a big issue, but special care in the presentation would be much appreciated.
> >
> > Also, I'd like to thank the authors to suggest many presentation changes. As always, you should take any of the reviewers' suggestions with a grain of salt, so no need to make changes the authors may consider unnatural. Yet, every attempt in improving the clarity of the paper is welcome.
> >
> > Finally, I maintain my opinion that this is a neat and good theoretical contribution.

---

> > > ### Author Response · Authors · 2022-08-05
> > > **Thanks for your kind reply!**
> > >
> > > Dear Reviewer m3LC,
> > >
> > >
> > > Thanks for your kind reply! All the reviews are very helpful, and we will improve our paper accordingly.
> > >
> > >
> > > Best
> > >
> > > Authors

---

### Official Review · Reviewer_CVs7 · 2022-07-11

**Rating:** 7
**Confidence:** 4
**Soundness:** 3 good
**Presentation:** 3 good
**Contribution:** 3 good

**Summary:**

In this paper, the authors propose a simple algorithm for smoothed online convex optimization (SOCO), which aims to minimize both the hitting cost and the switching cost. The key idea is to use a variant of Discounted-Normal-Predictor to combine different instances of online gradient descent (OGD). Based on the advantageous properties of Discounted-Normal-Predictor and OGD, they prove that the proposed algorithm attains nearly optimal (static and dynamic) regret bounds on every interval, even when the switching cost is present.

**Questions:**

1. In this paper, the authors choose the $\ell_2$-norm to measure the switching cost. Can the results be extended to other switching costs such as $\ell_1$-norm and $\ell_\infty $-norm?

2. The switching cost of $u_t$ is ignored in Theorem 4. Please explain the reason.

3. Currently, all the proofs are provided in the supplementary. It'd be better if the authors could highlight the sketch of analysis in the main content.

4. Competitive ratio is a popular performance measure for SOCO. The authors may need to explain why they have not used a competitive ratio.


**Strengths And Weaknesses:**

Strengths:
1. The writing of this paper is good, and the authors compare their bounds with previous results clearly. The proposed algorithm is the first attempt to simultaneously minimize adaptive regret and dynamic regret.

2. Although the analysis of Discounted-Normal-Predictor is quite involved, after we understand its property (Theorem 1) and usage (Lemma 1), it can be used as a black-box meta-algorithm, and the rest analysis is easy to follow. In this sense, the algorithm/analysis in this paper is simpler than that of Zhang et al. [2021c].

3. The theoretical guarantee of this paper is stronger than existing studies, since the proposed algorithm can minimize the dynamic regret with switching costs in every interval. For comparison, the dynamic regret bound of Zhang et al. [2021a] is non-adaptive, and the adaptive regret bound of Zhang et al. [2021c] only supports static regret.

Weaknesses:
1. As stated in the abstract, this paper is motivated by Kapralov and Panigrahy [2010] and Daniely and Mansour [2019], and thus the technical novelty is a bit limited. On the other hand, I do admit that the authors have made some modifications in Algorithms 2 and 3.

2. The authors have not discussed the competitive ratio of their algorithm.

Summary:
Although this paper is somehow incremental, the results are new and significant. It is still important to reveal the fact that Discounted-Normal-Predictor can be applied to SOCO and achieves better bounds, which may advance the theoretical and algorithmic developments of SOCO.

---

> ### Author Response · Authors · 2022-08-01
> **Many thanks for the constructive reviews!**
>
> Many thanks for the constructive reviews! We will revise our paper accordingly.
>
> ---
>
> Q1: Can the results be extended to other switching costs such as $\ell_1$-norm and $\ell_\infty$-norm?
>
> A1: It is possible to extend our results to $\ell_p$-norms based on the equivalence between vector norms, but the upper bound may depend on the dimensionality $d$. \
> For $\ell_1$-norm, we first have
> $$
> \sum _{t=r}^s  \big( f_t ( \mathbf{w}_t )+  \lambda G \\| \mathbf{w}_t  - \mathbf{w} _{t+1} \\| _1  \big) \leq \sum _{t=r}^s  \big( f_t ( \mathbf{w}_t )+  \lambda {\color{red} \sqrt{d}} G \\| \mathbf{w}_t  - \mathbf{w} _{t+1} \\| _{2}  \big) $$
> Then, from Theorem 4, we arrive at
> $$
> \sum _{t=r}^s  \big( f_t ( \mathbf{w}_t )+  \lambda G \\| \mathbf{w}_t  - \mathbf{w} _{t+1} \\| _1  \big) \leq  \sum _{t=r}^s  f_t(\mathbf{u}_t) + O\left( \sqrt{ (1+ \lambda {\color{red} \sqrt{d}} ) \tau (1+P _{r,s} ) \log T } \right)
> $$
> For $\ell_\infty$-norm, we first have
> $$
> \sum _{t=r}^s  \big( f_t ( \mathbf{w}_t )+  \lambda G \\| \mathbf{w}_t  - \mathbf{w} _{t+1} \\| _{\infty}  \big) \leq \sum _{t=r}^s  \big( f_t ( \mathbf{w}_t )+  \lambda G \\| \mathbf{w}_t  - \mathbf{w} _{t+1} \\| _{2}  \big)
> $$
> Then, based on Theorem 4, we obtain
> $$
> \sum _{t=r}^s  \big( f_t ( \mathbf{w}_t )+  \lambda G \\| \mathbf{w}_t  - \mathbf{w} _{t+1} \\| _{\infty}  \big) \leq \sum _{t=r}^s  f_t(\mathbf{u}_t) + O\left( \sqrt{ (1+ \lambda) \tau (1+P _{r,s} ) \log T } \right)
> $$
>
> ---
>
> Q2: The switching cost of $\mathbf{u}_t$ is ignored in Theorem 4. Please explain the reason.
>
> A2: Notice that the switching cost of $\mathbf{u}_t$  is  $P _{r,s}$, and
>
> $$
> P_{r,s} \leq \sqrt{D \tau P_{r,s} }  .
> $$
> As a result, subtracting $ P_{r,s} $ from the $O\left( \sqrt{ (1+ \lambda) \tau (1+P _{r,s} ) \log T } \right)$
> bound in Theorem 4 does not change its order, and thus we drop it for brevity.
>
> ---
>
> Q3: It'd be better if the authors could highlight the sketch of analysis in the main content.
>
> A3: Thanks for the suggestion. We will improve the writing and provide the sketch of the analysis in the final version.
>
> ---
>
> Q4: The authors may need to explain why they have not used a competitive ratio.
>
> A4: As explained in Line 100, all the studies on competitive ratio assume the *lookahead* setting, in which the learner first observes the hitting cost $f_t(\cdot)$ and then selects the decision $\mathbf{w}_t$. It is the lookahead ability makes it possible to derive constant competitive ratio. On the other hand, we follow the standard setting of OCO in which the learner *cannot* observe $f_t(\cdot)$ when submitting $\mathbf{w}_t$ [Gradu et al., 2020; Zhang et al., 2021c]. In this case, competitive ratio is unsuitable because it could be unbounded, and (dynamic) regret becomes the natural choice.

---

### Official Review · Reviewer_itGk · 2022-07-11

**Rating:** 7
**Confidence:** 3
**Soundness:** 4 excellent
**Presentation:** 4 excellent
**Contribution:** 4 excellent

**Summary:**

This paper proposes a new algorithm based on Discounted-Normal-Predictor to combine multiple experts for SOCO.  First, a modified Discounted-Normal-Predictor (DNP-cu) which uses a conservative update instead of the traditional projected update is proposed as the meta algorithm, followed by the prediction performance bound. Then a combiner based on DNP-cu is proposed to predict the combining weight for multi-expert SOCO. With OGD as the experts, the smoothes OGD in Algorithm 4 combines multiple experts by the proposed combiner. The authors give nearly optimal bounds for both the adaptive regret and dynamic regret for any lambda and any interval.

**Questions:**

The proposed expert combiner is really interesting and the theoretical results can recover many recent SOCO and even general online optimization works. I like this work while I have some minor questions:
1. Comparing with the expert combiner for SOCO in Zhang et al. [2021a], the dynamic regret bound of this paper is up to a logarithmic factor. Could the authors discuss more about the comparisons with the combiner in Zhang et al. [2021a]?
2. I noice that the dynamic regret bound nearly match the lower bound. Is it possible to derive a lower bound regret of the proposed algorithm to show what performance the algorithm can achieve?

**Limitations:**

The limitations are adequately addressed.

**Strengths And Weaknesses:**

**Originality \& Significance**

An effective design to better exploit multiple experts in online optimization is really important and desirable. The idea to combine the experts by predicting the combining weight based on DNP-cu is novel and very inspiring.

**Quality**

The regret analysis in this paper is solid. Both the dynamic regret and the more fundamental adaptive regret are bounded to match the state-of-the-art regret bound.

**clarity**

The paper is well-written and easy to follow.

Typos:  Line 532 in appendix, Theorem 6,  the output of OGD is $w_t$ in stead of $x_t$?

Line 551 in appendix, Eqn. (47), there should be a sum notation after the first "$\leq$".

---

> ### Author Response · Authors · 2022-08-01
> **Many thanks for the constructive reviews!**
>
> Many thanks for the constructive reviews! We will revise our paper accordingly.
>
> ---
>
> Q1: Comparing with the expert combiner for SOCO in Zhang et al. [2021a], the dynamic regret bound of this paper is up to a logarithmic factor. Could the authors discuss more about the comparisons with the combiner in Zhang et al. [2021a]?
>
> A1: The reason that Zhang et al. [2021a] do not suffer the $\log T$ factor is because they only consider the whole interval $[1,T]$. In contrast, our dynamic regret bound holds for *any* possible interval, and the $\log T$ factor is the price paid for adaptivity.\
> The combiner in Zhang et al. [2021a] is the standard Hedge algorithm, which is applied to the linearized loss with switching cost. It attains an $O(\sqrt{T})$ bound of regret over the interval $[1,T]$, but ignores all the other intervals (cf. Lemma 1 in their paper). On the other hand, our combiner is able to minimize the regret with switching cost in *every* interval (i.e., Corollary 2). If we only care about the single interval $[1,T]$, we can also get rid of the $\log T$ factor based on Lemma 15 of Kapralov and Panigrahy [2010].
>
> ---
>
> Q2: I notice that the dynamic regret bound nearly match the lower bound. Is it possible to derive a lower bound regret of the proposed algorithm to show what performance the algorithm can achieve?
>
> A2: Deriving the lower bound of a particular algorithm is really challenging, since we need to identify the hardest problem instance for that algorithm. Currently, we do not have a clear answer, and would like to leave it as a future work.
>
> ---
>
> Q3: Line 532 in appendix, Theorem 6, the output of OGD is $\mathbf{w}_t$ instead of $\mathbf{x}_t$?
>
> A3: Yes. It should be $\mathbf{w}_t$.
>
> ---
>
> Q4: Line 551 in appendix, Eqn. (47), there should be a sum notation after the first "$\leq$”.
>
> A4: Thanks. We will add the missing $\sum_{t=r}^s$.

---

### Official Review · Reviewer_ijEX · 2022-07-17

**Rating:** 5
**Confidence:** 2
**Soundness:** 3 good
**Presentation:** 3 good
**Contribution:** 3 good

**Summary:**

This paper studies smoothed online convex optimization. In particular, the authors proposed an online GD method with different step size predicted by the discounted normal predictor. The proposed combination is claimed to be simpler than previous literature while maintaining a similar regret bound.

**Questions:**

NA

**Strengths And Weaknesses:**

Strength: This paper is well written and easy to follow. The proposed algorithm is clear and simpler than the previous literature. Merging the DNP into the online gradient descent sounds novel to me.

Weakness:
1. No experiments are added to justify the advantage of the result.
2. It would be better demonstrated which role DHP is playing in the main algorithm to better highlight the contribution of this paper.

---

> ### Author Response · Authors · 2022-08-01
> **Many thanks for the constructive reviews!**
>
> Many thanks for the constructive reviews! We will revise our paper accordingly.
>
> ---
>
> Q1: No experiments are added to justify the advantage of the result.
>
> A1: Thanks for the suggestion. We would like to conduct experiments to verify our algorithm. On the other hand, we want to emphasize that this paper, as well as closely related works [Zhang et al., 2021a,c], is mainly theoretical. We have made the following theoretical contribution, which in our opinion could stand on its own.
> 1. Based on Discounted-Normal-Predictor, we propose a very simple algorithm for smoothed online convex optimization (SOCO).
> 2. This is the first effort to minimize both adaptive regret and dynamic regret under the setting of SOCO, and our results are stronger than existing ones [Zhang et al., 2021a,c].
> 3. We provide a novel analysis of Discounted-Normal-Predictor with conservative updating (DNP-cu), and prove that it suffers a small loss on every interval, even in the presence of switching costs.
>
> ---
>
> Q2: It would be better demonstrated which role DHP is playing in the main algorithm to better highlight the contribution of this paper.
>
> A2: We have explained the role of DNP-cu in the **Remark** after Corollary 2. Because there is a max operator in (13), we can derive two different lower bounds in (15) and (16), which can be used by Lemma 1 to bound the meta-regret. Based on (15) and (22), we can upper bound the regret of the meta-algorithm $\mathcal{A}$ w.r.t. the first algorithm $\mathcal{A}_1$. Similarly, from (16) and (23), we can upper bound the regret of $\mathcal{A}$ w.r.t. $\mathcal{A}_2$. When applying DNP-cu (i.e., Algorithm 3) to combine multiple algorithms sequentially, (15) and (16) play two different roles, as explained below.
> 1. Based on (15), we do not destroy the theoretical guarantee of early algorithms. Th reason is because the bound in (15) is *independent* of the interval length $\tau$, which is a very strong property. For details, please refer to (66) of the supplementary.
> 2. Based on (16), we can inherit the theoretical guarantee of the current algorithm. This is rather straightforward, and can be found in (65) of the supplementary.

---

### Meta-Review · Area_Chair_PSvs · 2022-08-23

**Recommendation:** Accept
**Confidence:** Certain

**Metareview:**

All the reviewers were happy with this paper. There were some comments about experiments and additional results (e.g. a lower bound), but the reviewers generally thought the work in the paper is solid enough to merit acceptance. I encourage the authors to incorporate the discussions that clarify various points in the final manuscript. It would also be nice to have some (at least toy) experiments to corroborate the theory.



**Award:**

No

---

### Decision · Program_Chairs · 2022-09-14

Accept